# Neural B-frame Video Compression with Bi-directional Reference Harmonization

**Yuxi Liu**[1,2*] **Dengchao Jin**[2] **Shuai Huo**[2†] **Jiawen Gu**[2‡]
**Chao Zhou**[2] **Huihui Bai**[3] **Ming Lu**[1†] **Zhan Ma**[1]

[1]Nanjing University, Nanjing, China   [2]Kuaishou Technology, Beijing, China
[3]Beijing Jiaotong University, Beijing, China
`yuxiliu@smail.nju.edu.cn`, {`jindengchao, huoshuai, gujiawen, zhouchao`}`@kuaishou.com`
`hhbai@bjtu.edu.cn`, {`minglu, mazhan`}`@nju.edu.cn`

## Abstract

Neural video compression (NVC) has made significant progress in recent years, while neural B-frame video compression (NBVC) remains underexplored compared to P-frame compression. NBVC can adopt bi-directional reference frames for better compression performance. However, NBVC's hierarchical coding may complicate continuous temporal prediction, especially at some hierarchical levels with a large frame span, which could cause the contribution of the two reference frames to be unbalanced. To optimize reference information utilization, we propose a novel NBVC method, termed **B**i-directional **R**eference **H**armonization **V**ideo **C**ompression (BRHVC), with the proposed Bi-directional Motion Converge (BMC) and Bi-directional Contextual Fusion (BCF). BMC converges multiple optical flows in motion compression, leading to more accurate motion compensation on a larger scale. Then BCF explicitly models the weights of reference contexts under the guidance of motion compensation accuracy. With more efficient motions and contexts, BRHVC can effectively harmonize bi-directional references. Experimental results indicate that our BRHVC outperforms previous state-of-the-art NVC methods, even surpassing the traditional coding, VTM-RA (under random access configuration), on the HEVC datasets. The source code is released at `https://github.com/kwai/NVC`.

## 1   Introduction

Lossy video compression is a cornerstone technology in the digital era, enabling efficient storage and transmission of overwhelming video information across modern communication systems. To support a wide range of compression scenarios, various video coding schemes have been developed [6, 10], with low-delay (LD) P-frame coding and random access (RA) B-frame coding being the most prominent. These two schemes are widely used in global video coding systems.

In LD mode, only the preceding frame is referenced during each coding process. In contrast, RA coding uses a bi-directional hierarchical coding structure, which naturally utilizes more reference information and achieves better compression performance than LD. So RA is widely used in scenarios where compression efficiency is more important than latency. The coding processes of LD and RA are shown in Figure 1, within a coding scenario with an Intra Period (IP) of 4 frames, an I-frame is

---

*This work was done when Yuxi Liu was a full-time intern at Kuaishou Technology.
†Corresponding author.
‡Project leader.

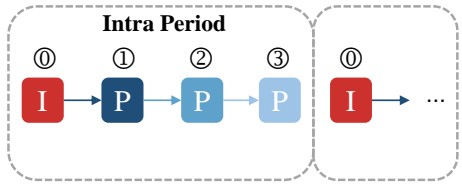 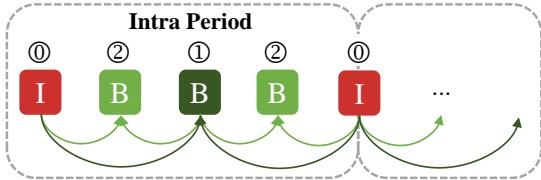

(a) Low-delay (LD) P-frame coding.   (b) Random access (RA) B-frame coding.

Figure 1: Two main coding schemes with Intra Period 4 as an example. Each square represents a frame, and the frames are temporally ordered from left to right. Arrows represent the reference pipeline. The serial numbers ① ② ③ indicate the coding order.

encoded independently without referencing any frame, a P-frame references the preceding frame, while a B-frame can reference both forward and backward frames.

Traditional video coding standards such as H.264/AVC [47], H.265/HEVC [42], and H.266/VVC [7] incorporate both LD and RA configurations. With the rise of deep learning, some end-to-end neural video compression (NVC) methods have emerged and gradually surpassed traditional standards under LD configurations. Most NVC research focuses on P-frame coding, such as [30, 27, 1, 28, 15, 41, 33, 44, 18, 43, 3] and the DCVC series [20, 38, 21–23, 36, 35, 17]. Although some neural B-frame video compression (NBVC) methods [8, 51, 39] also have made notable advancements, there remains a certain gap between these methods and VTM-RA (the reference software of H.266/VVC under RA configuration). Meanwhile, some issues related to NBVC have not yet been thoroughly investigated.

The hierarchical coding scheme of NBVC exhibits distinctive characteristics uncommon in other video processing tasks, and even completely different from neural P-frame video compression. To be specific, the span of each LD coding process is only one frame, thus the representation of motion is relatively easier. However, the frame span of RA is so large at the initial hierarchical levels (e.g., the coding of ① in Figure 1 (b)) that the motions become more unpredictable. In that case, we can not expect all the reference frames to be dependable. We define this phenomenon as unbalanced reference contribution (URC). For example, see Figure 2, the information on the number plate can only be referenced in $x_b$ because the movement of the racehorse caused the number plate to be obscured in the previous moment in $x_f$. So the model needs to pay more attention to the target region of $x_b$.

To optimize reference information utilization under the influence of the URC issue, we propose a novel NBVC method, termed **B**i-directional **R**eference **H**armonization **V**ideo **C**ompression (BRHVC), with the proposed Bi-directional Motion Converge (BMC) and Bi-directional Contextual Fusion (BCF). To expand the receptive field of the motion estimation, we downsample the reference frames and compute optical flows across multiple scales. Then, BMC converges compressed representations of the flows and enriches them. This method helps to alleviate the difficulty in estimating motion over large spans. After that, BRHVC conducts more accurate motion compensation on the reference contexts. Furthermore, the proposed BCF explicitly models the weights of reference contexts under the guidance of motion compensation accuracy. With more efficient motions and contexts, BRHVC can effectively harmonize bi-directional references. Experimental results indicate that our BRHVC outperforms previous state-of-the-art NVC methods, even surpassing VTM-RA on the HEVC datasets. The main contributions of this paper are summarized as follows:

- To cope with the unbalanced reference contribution between reference frames, we propose a novel NBVC method, termed BRHVC, with the proposed BMC and BCF.

- We propose Bi-directional Motion Converge (BMC) to converge multiple optical flows in motion compression, leading to more accurate motion compensation on a larger scale.

- We propose Bi-directional Contextual Fusion (BCF) to explicitly model the weights of reference contexts under the guidance of motion compensation accuracy, which facilitates the harmonization of bi-directional references.

- Experiment results show that our BRHVC outperforms previous SOTA NVC methods and surpasses VTM-RA on the HEVC datasets.

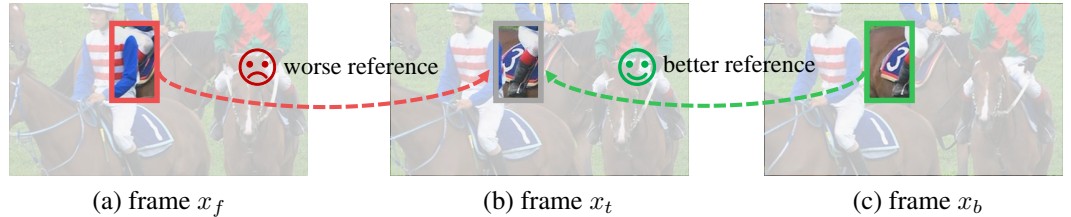

(a) frame $x_f$         (b) frame $x_t$         (c) frame $x_b$

Figure 2: The unbalanced contribution issue between reference frames in B-frame coding. The right reference is notably more significant than the left for the compression of the number plate.

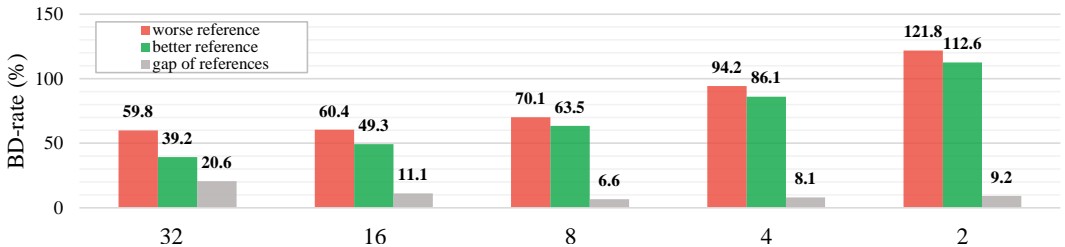

Figure 3: Quantitative experiment on unbalanced reference contribution. The results of the "gap of references" indicate the average contribution difference between two reference frames across the frame spans of {32, 16, 8, 4, 2}. Refer to Section 3 for more explanations.

## 2 Related Work

### 2.1 Neural P-frame Video Compression

Recently, many methods have adopted neural network (NN) to improve video coding, including using NN to enhance traditional video coding [32, 10, 16, 55, 12, 26, 24] and NN-based end-to-end video coding. Earlier end-to-end neural P-frame video compression (NPVC) methods still follow the traditional predictive coding structure [30, 31, 27, 1, 28, 29, 14, 15, 41, 25]. For example, Lu et al. [30, 31] proposed the first NPVC method, termed DVC. This framework generates and transmits the motion vectors to get the predicted frame. Then the residual between the predicted and current frames is compressed to achieve bitrate savings. However, residuals in the pixel domain lack sufficient information to fully leverage the representational capabilities of neural networks. To address it, Li et al. [20] proposed a deep contextual video compression framework, termed DCVC, which enables a paradigm shift from predictive coding to conditional coding. Based on DCVC, Sheng et al. [38] proposed a temporal context mining (TCM) module to learn richer and more accurate contexts at multi-scales. This multi-scale contextual compression paradigm has been employed in many methods. Among them, DCVC-DC [22] and DCVC-FM [23] have achieved state-of-the-art (SOTA) performance, saving approximately 20% of the bitrate compared to the LDB mode of VTM-17.0.

### 2.2 Neural B-frame Video Compression

Neural B-frame video compression approaches can also be broadly summarized in two paradigms, predictive-based [48, 11, 52, 54, 2, 49, 8, 16] and condition-based [51, 39]. Predictive-based methods compute the predicted frame as the primary reference, with the help of two reference frames. Some methods [48, 11, 2, 49] turn to an interpolation network with a view to avoiding motion compression. These interpolation networks often assign a binary mask to each prediction frame from two reference frames, indicating which pixels are more accurate than the other. Since it shows some degree of adaptation to the URC mentioned above, some other predictive-based works [52, 54, 8, 16] also introduce the binary mask to obtain more accurate estimation at the pixel domain. However, as discussed in Section 2.1, condition-based methods [51, 39] have high-dimension multi-scale contextual references in the feature domain, better than the pixel domain of predictive-based methods. For example, Yang et al. [51] proposed a unified contextual video compression framework for joint P-frame and B-frame coding. Sheng et al. [39] proposed an efficient model, DCVC-B, with a customized training strategy, achieving SOTA performance in NBVC.

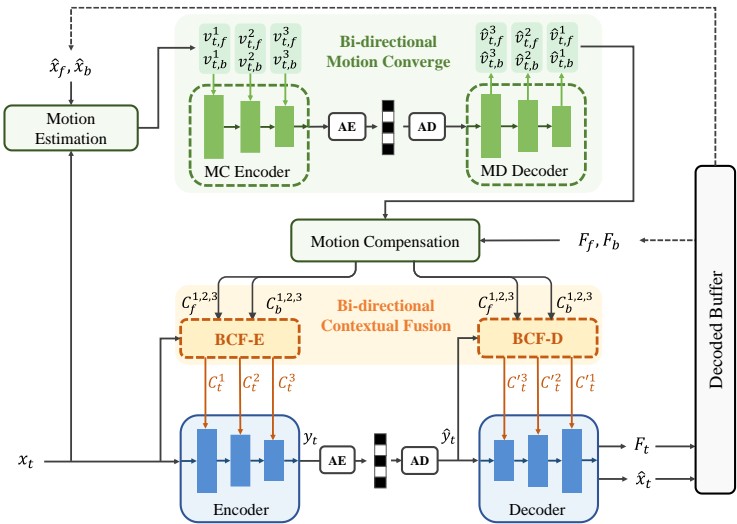

Figure 4: The overall architecture of BRHVC. AE and AD denote arithmetic encoding and arithmetic decoding. MC Encoder and MD Decoder denote Motion Converge Encoder and Motion Diverge Decoder. $C_f^{1,2,3}$ and $C_b^{1,2,3}$ denote $\{C_f^1, C_f^2, C_f^3\}$ and $\{C_b^1, C_b^2, C_b^3\}$, respectively. We omit the entropy model and some outputs from the decoded buffer for brevity.

However, the above condition-based methods do not design a specific module for the unbalanced contribution issue, merely concatenating the contexts from the two reference frames as the primary reference. Whether it can adequately model the weights of the reference information remains an open question. It will be discussed and addressed in this paper.

## 3 Rethink to Unbalanced Reference Contribution

To investigate how much the URC affects compression performance, we developed an NBVC baseline model (the details will be introduced later) and conducted a quantitative experiment using it. For each video sequence, we only compressed the 16th frame $x_{16}$, setting the two reference frames $\{\hat{x}_f, \hat{x}_b\}$ to $\{\hat{x}_0, \hat{x}_{32}\}, \{\hat{x}_8, \hat{x}_{24}\}, \{\hat{x}_{12}, \hat{x}_{20}\}, \{\hat{x}_{14}, \hat{x}_{18}\}, \{\hat{x}_{15}, \hat{x}_{17}\}$, corresponding to spans of 32, 16, 8, 4 and 2. In each compression, we only use a single reference like $\{\hat{x}_f, \hat{x}_f\}$ or $\{\hat{x}_b, \hat{x}_b\}$, then calculate their BD-rates [4] with referring $\{\hat{x}_f, \hat{x}_b\}$ (the default setting) as the anchor. The lower one is taken as "better reference", while the higher one is "worse reference". We then statistically average the BD-rate results over the HEVC datasets [5] as shown in Figure 3. It can be seen that with the large span, the gap between the better reference and the worse reference is substantial, reaching 20.6% and 11.1% for spans of 32 and 16, respectively. As the span decreases, the BD-rate for both references increases dramatically. It is because with the small span, the reference information is more relevant to the target, making both of them more important. Therefore, the absence of any reference leads to severe degradation. Considering the significant impact of the URC issue in NBVC, there should be a method that can effectively determine the weights of the two references to better harmonize the reference information. More details of this quantitative experiment are in Appendix A.1.

## 4 Method

### 4.1 The Overall Framework

We build our baseline on the backbone of DCVC-B [39] and DCVC-DC [22], while some of the modules have certain differences. More details of our network design are in Appendix A.2. Our BRHVC introduces the proposed BMC and BCF based on the baseline. As shown in Figure 4, given the current frame $x_t$ to be encoded, we use the two decoded reference frames $\hat{x}_f, \hat{x}_b$ and their features $F_f, F_b$ as reference. When the IP length is 32, the frame span $b - f$ equals $2^{6-level}$, where $level$ denotes the heirachical level of $x_t$. The overall process of one frame coding is as follows:

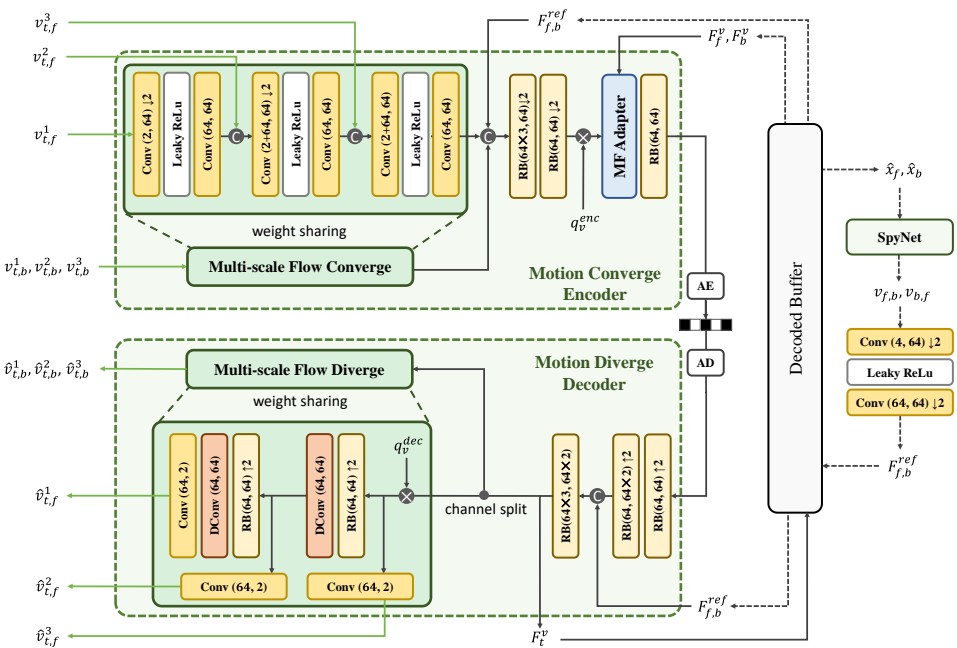

Figure 5: The framework of Bi-directional Motion Converge. MF Adapter denotes Motion Feature Adapter. $q_v^{enc}$ and $q_v^{dec}$ are learnable quantization vectors for variable bitrates [22].

**Motion Estimation and Compression.** We use Spynet [37] to generate multi-scale optical flows $\{v_{t,f}^1, v_{t,f}^2, v_{t,f}^3\}$ and $\{v_{t,b}^1, v_{t,b}^2, v_{t,b}^3\}$ from two directions. To be specific, we downsample the original image $x_t$ and reference images $\hat{x}_f, \hat{x}_b$ twice, then separately compute optical flows by SpyNet at each scale. This approach facilitates the expansion of the receptive field for motion estimation, thereby accommodating larger frame spans. Then we use the proposed BMC to compress and reconstruct these optical flows. Refer to Section 4.2 for more details.

**Motion Compensation.** Given the reconstructed optical flows $\{\hat{v}_{t,f}^1, \hat{v}_{t,f}^2, \hat{v}_{t,f}^3\}$ and $\{\hat{v}_{t,b}^1, \hat{v}_{t,b}^2, \hat{v}_{t,b}^3\}$, we use them to warp the reference features at each scale. During this process, $F_f, F_b$ individually pass through a share-weighted contextual-feature extraction [22, 39] (denoted as $CFE$, which combines Temporal Context Mining [20, 32] with Group-Based Offset Diversity [20]) with the flows, resulting in features $\{C_f^1, C_f^2, C_f^3\}$ and $\{C_b^1, C_b^2, C_b^3\}$ at multi-scales, where $C_f^1, C_b^1 \in \mathbb{R}^{48 \times H \times W}$, $C_f^2, C_b^2 \in \mathbb{R}^{64 \times H/2 \times W/2}$, and $C_f^3, C_b^3 \in \mathbb{R}^{96 \times H/4 \times W/4}$. It can be expressed as:

$$C_f^1, C_f^2, C_f^3 = CFE(F_f, \hat{v}_{t,f}^1, \hat{v}_{t,f}^2, \hat{v}_{t,f}^3), \quad C_b^1, C_b^2, C_b^3 = CFE(F_b, \hat{v}_{t,b}^1, \hat{v}_{t,b}^2, \hat{v}_{t,b}^3). \quad (1)$$

**Transform and Entropy Coding.** NVC methods use a neural network-based encoder and decoder for coding's transform to reduce the dimension of the latent variables. The proposed BCF outputs the multi-scale contexts $\{C_t^1, C_t^2, C_t^3\}$ for encoder and $\{C_t'^1, C_t'^2, C_t'^3\}$ for decoder (see more details in Section 4.3) as refined reference contexts. After the transform, arithmetic coding is utilized to compress latent variables $y_t$ into a bitstream with the probability estimated by an entropy model. Our BRHVC adopts the quadtree partition-based entropy model [22].

## 4.2 Bi-directional Motion Converge

As the frame span increases, the motions will become larger and more complex. However, many current motion estimation networks are designed for a frame span of one. To capture large motion information as much as possible, a straightforward idea is to downsample the input frames multiple times, so that the network can obtain a larger receptive field. Nevertheless, this approach increases the information content of the optical flows, requiring a customized compression method.

To achieve more efficient optical flow compression, previous works [39, 53, 52] employ a bi-directional motion residual compression (denoted as BMRC in this paper), which estimates the bi-

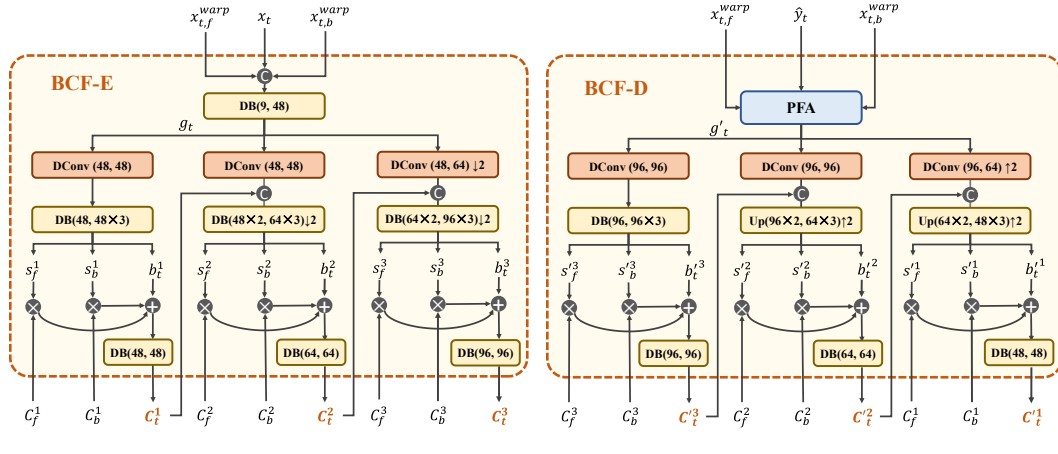

(a) BCF module for encoder (BCF-E)     (b) BCF module for decoder (BCF-D)

Figure 6: The network architecture of Bi-directional Contextual Fusion (BCF) for encoder and decoder. PFA denotes Pixel Feature Alignment module.

directional flows $v_{b,f}, v_{f,b}$ between two reference frames and compresses the residuals $(v_{t,b}-0.5 \cdot v_{f,b})$ and $(v_{t,f} - 0.5 \cdot v_{b,f})$ to eliminate redundancy. Our baseline model uses BMRC for subsequent comparisons. However, BMRC is not suitable for our multi-scale optical flows for two main reasons: first, computing residuals across multiple scales is overly complicated; and second, residuals in the optical flow domain, like those in the pixel domain, are not optimal as mentioned in Section 2.1.

To effectively compress the multi-scale flows, we propose Bi-directional Motion Converge. The framework of BMC is shown in Figure 5. In optical flow compression, BMC converges multi-scale optical flows into a single latent variable and fuses the latent variables from the two directions. After compression, these optical flows diverge through the reverse process and are used for motion compensation in the next part. On this basis, BMC introduces the flow features of two directions from the decoded buffer, $\{F_f^v, F_b^v\}$ and $F_{f,b}^{ref}$ as priors. Motion Feature Adapter selects different sub-networks to adapt prior information $\{F_f^v, F_b^v\}$ based on the frame type (B-frame or I-frame), as detailed in Appendix A.2. $\{F_f^v, F_b^v\}$ provides the motion of two references individually, which is from the hierarchical layer with a larger span. In contrast, $F_{f,b}^{ref}$ is the feature-domain representation of $v_{f,b}$ and $v_{b,f}$, providing the motion information between two reference frames, spanning the flows to be compressed (i.e., $v_{t,b}, v_{t,f}$). This approach enriches prior information compared to the residual-based approach BMRC. Unlike previous works [39, 20, 38, 22, 23], which obtain small-scale flows through downsampling, our BMC employs specific optical flows at different scales, thereby increasing the diversity of motion representation.

### 4.3 Bi-directional Contextual Fusion

In previous works [39, 51], the contexts $\{C_f^1, C_f^2, C_f^3\}$ and $\{C_b^1, C_b^2, C_b^3\}$ are fused with the latent features of the current frame at different scales in the encoder and decoder. Specifically, the features of the current frame and the two references are concatenated along the channel dimension and then fused through subsequent convolution layers. However, this concatenation design may assume that the two reference frames have equal weights, i.e., they are considered to be of equal importance. In reality, the contributions of the two reference frames are often different, as discussed in Section 3. It is more common in large frame spans, as the motion becomes more unpredictable. Even in some cases, a region of the current frame may not benefit from either reference frame, which requires reducing the weights of both references and increasing the weight of the current frame's own features.

Based on the above motivation, we propose the Bi-directional Contextual Fusion (BCF) that can adaptively adjust the weights of the reference frames for each region. Its structure is shown in Figure 6, where DConv denotes Depthwise Separable Convolution [9], DB denotes Depth Block [22]. UP denotes upsampling using the pixel shuffle [40] convolution kernel. All convolutions there have

a default kernel size of 3×3. PFA denotes the proposed Pixel Feature Alignment module, and the specific module architectures can be seen in Appendix A.2.

First of all, BCF analyses the similarity between the current frame and the two motion-compensated reference frames as the guidance $g_t$. The three pixel-domain images $\{x_t, x_{t,f}^{warp}, x_{t,b}^{warp}\}$ are used for computing the guidance to fuse $\{C_f^1, C_b^1\}$, $\{C_f^2, C_b^2\}$, and $\{C_f^3, C_b^3\}$. The process is as follows:

$$
\begin{aligned}
C_t^1, C_t^2, C_t^3 &= BCF_E(\{C_f^1, C_b^1\}, \{C_f^2, C_b^2\}, \{C_f^3, C_b^3\} | \{x_t, x_{t,f}^{warp}, x_{t,b}^{warp}\}), \\
where \quad x_{t,f}^{warp} &= warp(\hat{x}_f, \hat{v}_{t,f}^1), \quad x_{t,b}^{warp} = warp(\hat{x}_b, \hat{v}_{t,b}^1),
\end{aligned}
\tag{2}
$$

$BCF_E$ represents BCF for the encoder (BCF-E). BCF-E generates reference weight information based on the differences from the pixel domain, and through multi-scale weight information propagation, each layer obtains adaptive weight allocation. To avoid the situation where the context features of both reference frames cannot provide accurate reference information, $b_t^i$ provides an independent supplementary bias. The weights and the bias are denoted as $s_f^i, s_b^i$ and $b_t^i$, their dimensions are the same as those of $C_f^i, C_b^i$, where $i$ represents the scale. We denote the depth block as $DB_i$. $\otimes$ means element-wise multiplication. The specific formulation of $C_t^1, C_t^2, C_t^3$ can be represented as follows:

$$
\begin{aligned}
C_t^1 &= DB_1(s_f^1 \otimes C_f^1 + s_b^1 \otimes C_b^1 + b_t^1), \\
C_t^2 &= DB_2(s_f^2 \otimes C_f^2 + s_b^2 \otimes C_b^2 + b_t^2), \\
C_t^3 &= DB_3(s_f^3 \otimes C_f^3 + s_b^3 \otimes C_b^3 + b_t^3).
\end{aligned}
\tag{3}
$$

Besides, after decoding the bitstream to obtain $\hat{y}_t$ at the decoder, the decoder will use the BCF-D for decoding. The main difference between BCF-E and BCF-D is that the guidance information $g_t$ and $g_t'$ are from different sources. $x_t$ is not available to the decoder, so we utilize $\hat{y}_t$ with the PFA module for alignment between pixels and features. The decoding process is as follows:

$$
C_t'^1, C_t'^2, C_t'^3 = BCF_D(\{C_f^1, C_b^1\}, \{C_f^2, C_b^2\}, \{C_f^3, C_b^3\} | \{\hat{y}_t, x_{t,f}^{warp}, x_{t,b}^{warp}\}).
\tag{4}
$$

After harmonizing the reference weights, these contexts are concatenated with the latent features in the encoder and decoder at three scales. It could support more accurate reference information for transform, achieving more bitrate saving and reconstruction performance.

## 5 Experiments

### 5.1 Settings

**Training.** We use Vimeo-90k [50] to train BRHVC from scratch with 7-frame sequences. Then we fine-tune BRHVC on original Vimeo videos with 17-frame sequences following [39, 23]. We use the multi-stage training strategy in [39]. The video frames are randomly cropped into 256×256 patches. We randomly reverse the order of sequences with a probability of 50% as data augmentation. AdamW [19] is used as the optimizer with a batch size of 8.

**Testing.** We evaluate the compression performance on HEVC Class B~E [5], UVG [34], and MCL-JCV [46]. The resolution of the HEVC Class C~E is in the range of 240p to 720p, while HEVC Class B, MCJ-JCV, and UVG have a resolution of 1080p. The first 97 frames of all the datasets are used with Intral Period 32, i.e., coding 93 B/P-frames with 4 I-frames. We convert YUV420 inputs to RGB as the original sequences using BT.709 for all the methods. To compare the performance of traditional methods, we used the LDB and RA modes of the traditional standards software HM-16.5 [42, 13] and VTM-17.0 [7, 45]. Since traditional methods do not support compression of RGB format, following the practice in [22], traditional methods compress the sequences through YUV444 format and calculate the PSNR after converting back to RGB using BT.709. As for NVC methods, we compare the previous SOTA NPVC methods, DCVC-DC [22] and DCVC-FM [23]. In addition, we compare the SOTA NBVC method, DCVC-B [39], which has a similar structure to our baseline. We use the same I-frame model as DCVC-DC and DCVC-B for our coding. More details about the test configuration can be seen in Appendix A.3.

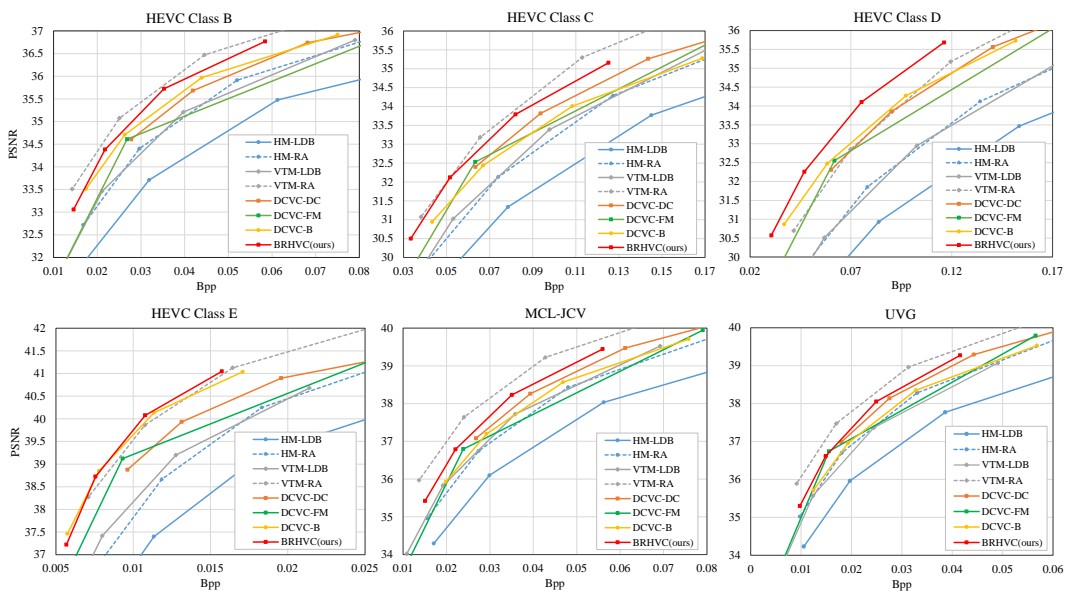

Figure 7: Rate-distortion curves for HEVC Class B~E, MCJ-JCV and UVG datasets.

Table 1: BD-rate (%) comparison for PSNR. **Black bold** indicates the best results within all methods, while **__underline bold__** indicates the best results within NVC methods.

|  | HEVC Class B | HEVC Class C | HEVC Class D | HEVC Class E | HEVC Average | MCL-JCV | UVG | All Average |
|---|---|---|---|---|---|---|---|---|
| VTM-LDB | 0.0 | 0.0 | 0.0 | 0.0 | 0.0 | 0.0 | 0.0 | 0.0 |
| VTM-RA | **-33.6** | **-29.1** | -30.2 | -30.7 | -30.9 | **-31.0** | **-33.5** | **-31.3** |
| HM-LDB | 38.5 | 35.4 | 32.5 | 42.3 | 37.1 | 41.9 | 36.3 | 37.8 |
| HM-RA | 0.2 | 4.2 | -0.7 | 7.1 | 2.7 | 5.4 | -4.9 | 1.8 |
| DCVC-DC | -12.4 | -12.5 | -28.4 | -18.2 | -17.8 | -8.9 | -16.8 | -16.2 |
| DCVC-FM | -15.2 | -19.2 | -34.2 | -26.5 | -23.7 | -8.3 | -18.9 | -20.3 |
| DCVC-B | -18.7 | -8.6 | -33.5 | -32.4 | -23.3 | -2.0 | -7.8 | -17.2 |
| BRHVC | **__-25.5__** | **__-25.2__** | **-44.7** | **-32.6** | **-32.0** | **__-16.1__** | **__-19.7__** | **__-27.3__** |

## 5.2 Comparison Results

The rate-distortion curves of all methods can be seen in Figure 7, and the BD-rate comparison is shown in Table 1. Our proposed BRHVC achieves an average bitrate saving of 27.3% compared to VTM-LDB. It outperforms the previous SOTA NVC methods (i.e., P-frame method DCVC-FM and B-frame method DCVC-B) on all the datasets. In the average performance on HEVC datasets, BRHVC achieves the best bitrate saving of 32.0%, higher than VTM-RA's 30.9%. Particularly in HEVC Class D, BRHVC can save 44.7% bitrate, which is significantly higher than other methods. These experimental results indicate that our method achieves a significant advancement in the NBVC domain. Additionally, some visual comparison results are shown in Appendix A.4.

## 5.3 Ablation Study and Complexity

To verify the validity of our proposed BCF and BMC, we performed ablation experiments as shown in Table 2. DCVC-B(re-trained) denotes that we re-train the open-source DCVC-B model from scratch with our settings. Our baseline and DCVC-B performances are close, suggesting that our subsequent improvement mainly comes from the proposed modules rather than the training settings. Due to the slightly different network design, our baseline demonstrates a slightly slower speed but with less GPU memory overhead. The results show that BCF can achieve a 6.6% bitrate saving, effectively enhancing reference information utilization to deal with URC issue. Additionally, the incorporation of BMC strengthens BRHVC's ability to represent large span motion, providing more

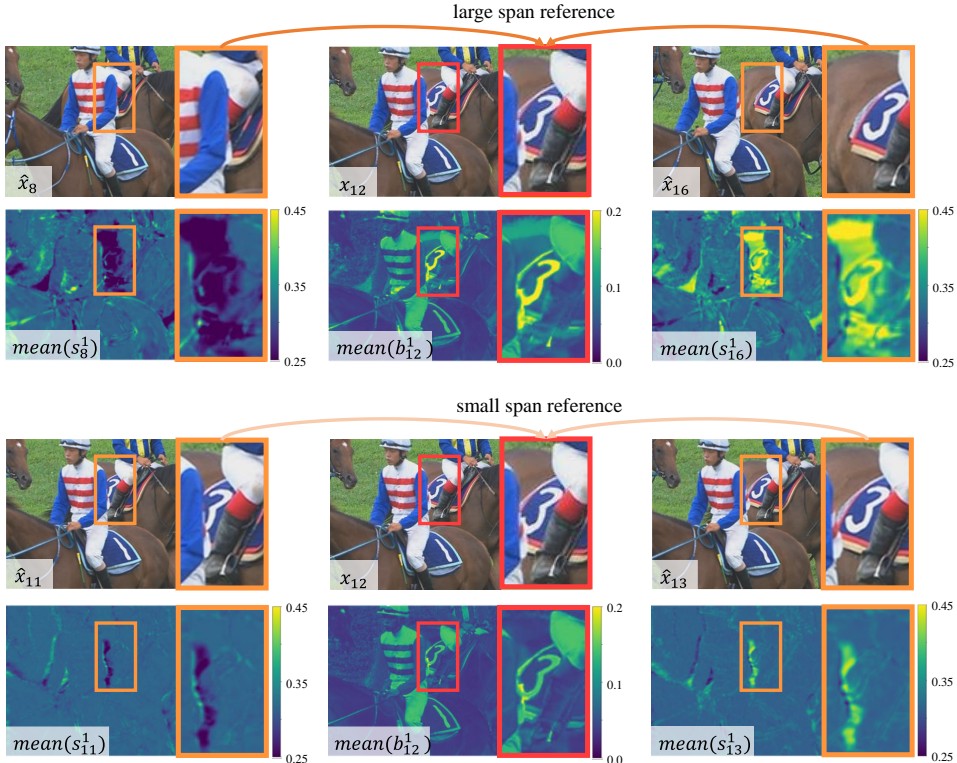

Figure 8: Visualization of BRHVC's reference harmonization effect on different spans. We compress $x_{12}$ and use $\hat{x}_8, \hat{x}_{16}$ (large span) or $\hat{x}_{11}, \hat{x}_{13}$ (small span) as references. We visualize the outputs $s_f^1, s_b^1, b_t^1$ from BCF-E. $mean(\cdot)$ denotes taking the absolute value and averaging over the channels.

Table 2: Ablation study on different network designs with the corresponding computation cost. The BD-rate results are calculated on HEVC datasets. We compare average single-frame codec times tested in 1080p sequences on one Nvidia RTX 4090 GPU. BL denotes our baseline model.

| Model | BD-rate | Parameters | kMACs/pixel | Memory | Enc. / Dec. time |
|---|---|---|---|---|---|
| DCVC-B(re-trained) | 0.4% | 24.4 M | 2934.9 | 13.5 G | 646 / 511 ms |
| BL | 0.0 | 23.7 M | 2831.9 | 9.6 G | 681 / 538 ms |
| BL w/ BMC | -6.4% | 24.3 M | 3170.9 | 10.6 G | 780 / 602 ms |
| BL w/ BCF | -6.6% | 28.9 M | 3532.8 | 11.4 G | 896 / 583 ms |
| BL w/ BCF, BMC | -12.3% | 29.4 M | 3887.8 | 12.4 G | 983 / 670 ms |

accurate reference to BCF, which further improves the performance to 12.3% bitrate saving with 24% additional decoding time. It can be observed that the additional computational overhead incurred is relatively acceptable compared to the significant improvements by our BRHVC.

## 5.4 Visulization

Figure 8 shows the visualization of our BRHVC to harmonize the references of different spans. In this example, the number plate is the focal target of our compression. The motions are more complex over large spans (i.e., $\hat{x}_8, \hat{x}_{16}$ as references), so the network needs to pay more attention to the useful one (i.e., $\hat{x}_{16}$). Thus, $mean(s_{16}^1)$ (computed by BCF-E) demonstrates a higher weight than $mean(s_8^1)$. On this basis, $mean(b_{12}^1)$ provides supplementary details to compensate for the loss of information. If the span is small, the difference between $mean(b_{11}^1)$ and $mean(b_{13}^1)$ will decrease because their information is both useful in most regions, and the supplementary $mean(b_{12}^1)$ will also decrease accordingly. It shows that our proposed modules are working as expected.

# 6 Conclusion and Limitation

In this paper, we introduce the phenomenon of unbalanced reference contribution in P-frame coding and analyze it quantitatively. To cope with it, we propose a novel NBVC method, termed BRHVC, with proposed BMC and BCF. BMC converges multiple optical flows in motion compression, leading to more accurate motion compensation for both references. BCF explicitly models the weights of reference contexts under the guidance of motion compensation accuracy, which facilitates the harmonization of bi-directional references. Experiment results show that our BRHVC outperforms previous SOTA NVC methods and surpasses VTM-RA on the HEVC datasets. However, our BRHVC does not perform sufficiently well on high frame rate datasets, such as MCL-JCV and UVG, which may require further investigation in future work.

# 7 Acknowledgements

The authors would like to express their sincere gratitude to the Kuaishou Research and Development Department (R&D), and the Interdisciplinary Research Center for Future Intelligent Chips (Chip-X) and Yachen Foundation for their invaluable support. This work was supported in part by National Key Research and Development Project of China (Grant No. 2022YFF0902402), Kuaishou Research and Development Department (R&D), Natural Science Foundation of Jiangsu Province (Grant No. BK20241226), Beijing Natural Science Foundation (L223022), and National Natural Science Foundation of China (Grant No. 62401251, 62431011, 62331003).

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

# A Technical Appendices and Supplementary Material

## A.1 Quantitative Experiment for the URC issue

**Experimental Setting.** For each video sequence, we only compressed the 16th frame $x_{16}$, setting the two reference frames $\{\hat{x}_f, \hat{x}_b\}$ to $\{\hat{x}_0, \hat{x}_{32}\}, \{\hat{x}_8, \hat{x}_{24}\}, \{\hat{x}_{12}, \hat{x}_{20}\}, \{\hat{x}_{14}, \hat{x}_{18}\}, \{\hat{x}_{15}, \hat{x}_{17}\}$, corresponding to spans of 32, 16, 8, 4 and 2. We configured the baseline model in three different settings: both reference frames are forward frames (denoted as $M(x_t|\hat{x}_f, \hat{x}_f)$, or $M_{f,f}$ for simplicity), both reference frames are backward frames (denoted as $M(x_t|\hat{x}_b, \hat{x}_b)$ or $M_{b,b}$), and the two reference frames are a forward frame and a backward frame (denoted as $M(x_t|\hat{x}_f, \hat{x}_b)$ or $M_{f,b}$, the default setting in B-frame compression), respectively. All the reference frames are compressed by the same I-frame model [22]. In each compression, the baseline model outputs four compressed results at four different bitrates, in order to calculate the BD-rate in one frame. Then we calculate the BD-rate of $M_{f,f}$ and $M_{b,b}$, respectively, with $M_{f,b}$ as the anchor. The lower one is taken as "better reference", while the higher one is "worse reference". We then statistically average the BD-rate results over the HEVC datasets as shown in Figure 3 and Table 3. The formulation is as follows:

$$\text{Better}(x_t|b - f) = min[BDrate^{M_{f,b}}(M_{f,f}), BDrate^{M_{f,b}}(M_{b,b})],$$
$$\text{Worse}(x_t|b - f) = max[BDrate^{M_{f,b}}(M_{f,f}), BDrate^{M_{f,b}}(M_{b,b})], \quad (5)$$

where $b - f$ means the span, $BDrate^{M_1}(M_2)$ denotes calculating the BD-rate of $M_2$ using $M_1$ as the anchor. $min[\cdot, \cdot]$ and $max[\cdot, \cdot]$ denotes taking the lower value and the higher value, respectively.

**Experimental Analysis.** The detailed results of the quantitative experiment in Section 3 is shown as Table 3. As mentioned in Section 3, with the large frame span, the gap between the better and the worse reference is substantial, reaching 20.6% and 11.1% for spans of 32 and 16, respectively. As the span decreases, the BD-rate for both references increases dramatically. It is because with the small span, the reference information is more relevant to the target, making both of them more important. Therefore, the absence of any reference leads to severe degradation. However, it should be noted that while the ratio of the gap (between "Better" and "Worse") to "Worse" exhibits an approximately monotonic decreasing trend with span, the value of the gap itself does not. This is also because, as the span decreases, the BD-rate of both "Worse" and "Better" increases, possibly causing the gap to increase correspondingly. Additionally, it can be observed that the gap is more pronounced on Class C and D. This may be due to the presence of many intense-motion scenes in these datasets, which increase frame differences and thus the gap.

Table 3: Quantitative experiment on the URC issue with detailed results. The metric is BD-rate(%).

| Dataset | Reference | Span of the Reference Frames | | | | |
|---|---|---|---|---|---|---|
| | | 32 | 16 | 8 | 4 | 2 |
| Average | Better | 39.2 | 49.3 | 63.5 | 86.1 | 112.6 |
| | Worse | 59.8 | 60.4 | 70.1 | 94.2 | 121.8 |
| Class B | Better | 32.2 | 39.1 | 53.3 | 76.4 | 114.1 |
| | Worse | 40.2 | 45.7 | 57.1 | 86.6 | 123.4 |
| Class C | Better | 33.6 | 46.7 | 53.4 | 63.7 | 83.8 |
| | Worse | 69.3 | 62.8 | 61.8 | 69.1 | 92.8 |
| Class D | Better | 39.4 | 50.9 | 72.1 | 113.2 | 148.3 |
| | Worse | 68.7 | 67.1 | 80.0 | 121.4 | 160.8 |
| Class E | Better | 58.3 | 68.2 | 83.1 | 96.1 | 101.1 |
| | Worse | 68.2 | 73.0 | 89.5 | 103.2 | 105.9 |

## A.2 Network Designs

**Encoder and Decoder.** As shown in Figure 9, we use the same architecture of the encoder and decoder as DCVC-DC [22]. The main difference between our architecture and DCVC-DC's is that we use the bi-directional reference contexts $\{C_t^1, C_t^2, C_t^3\}$ and $\{C'^1_t, C'^2_t, C'^3_t\}$ as contextual features, while the features of DCVC-DC are from a single preceding direction.

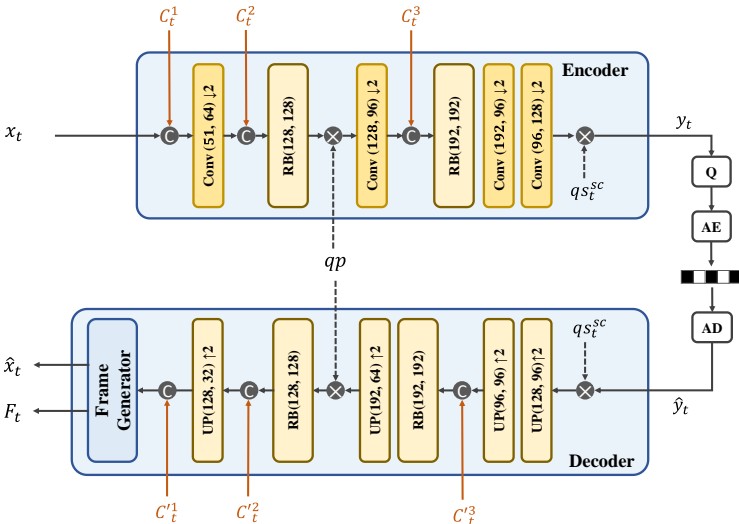

Figure 9: The network architecture of the encoder and decoder. $qp$ and $qs_t^{sc}$ are quantization variables for variable bitrates. "RB($C_{in}$,$C_{out}$)" is Residual Block with input channel $C_{in}$ and ouput channel $C_{out}$. Refer to [22] for more details.

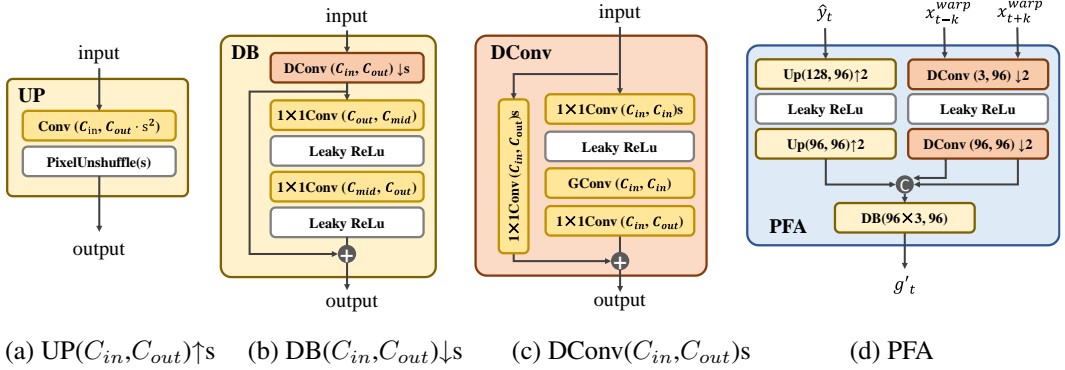

(a) UP($C_{in}$,$C_{out}$)↑s    (b) DB($C_{in}$,$C_{out}$)↓s    (c) DConv($C_{in}$,$C_{out}$)s    (d) PFA

Figure 10: The architecture of the sub-modules in BCF.

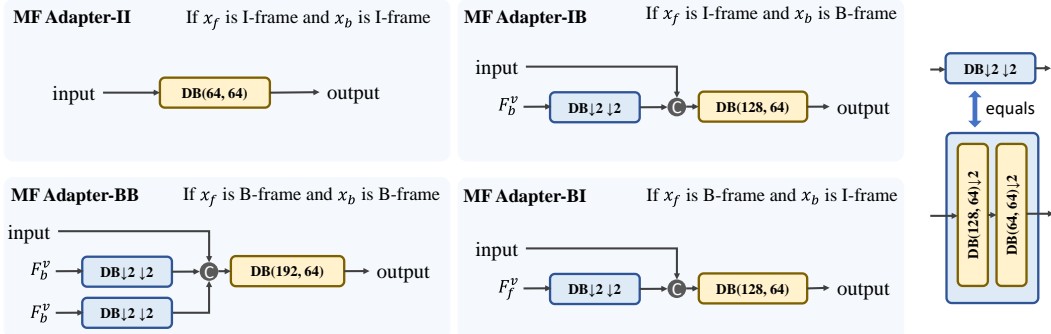

Figure 11: The architecture of Motion Feature Adapter (MF Adapter) in BMC. All the "DB↓2↓2" share the same network weights.

**Sub-Modules in BCF.** The detailed module designs of BCF is as shown in Figure 10. Not that there may be some parameter differences based on the basic architectures. For example, "DConv($C_{in}$,$C_{out}$)s" means DConv layer with input channel $C_{in}$ and ouput channel $C_{out}$, and "s" denotes the sample stride (defaults to 1). "↓s" means the downsample stride is "s", while "↑s" means upsample with stride "s".

**MF Adapter in BMC.** Motion Feature Adapter selects different sub-networks to adapt prior information $\{F_f^v, F_b^v\}$ based on the frame type (B-frame or I-frame). For example, if the reference frame $x_f$ is B-frame, the I-frame model does not need to compute the optical flow and therefore can not output $F_f^v$ as an intermediate result. In that case, we can not introduce $F_f^v$. If both frames are I-frames, the input is solely processed to generate the output (see "MF Adapter-II" in Figure 11).

## A.3 Test Configuration

We compare traditional methods HM-16.5 and VTM-17.0 with LDB and RA configuration in Section 5.1. To evaluate the compression performance of traditional methods, we follow the configuration of [22] for fair comparison. We convert YUV420 inputs to RGB as the original sequences using BT.709 for all the methods. Since traditional methods do not support compression of RGB format, traditional methods compress the sequences through YUV444 format and calculate the PSNR after converting back to RGB using BT.709. To configure the best traditional codecs, we use the 10 bit depth for YUV444 colorspace. These methods share some command-line arguments. While for the {*config_file*}, VTM-RA sets it to *encoder_randomaccess_vtm.cfg*, VTM-LDB sets it to *encoder_lowdelay_vtm.cfg*, HM-LDB sets it to *encoder_lowdelay_main_rext.cfg*, and HM-RA sets it to *encoder_randomaccess_main_rext.cfg*. The command line is as follows:

- -c {*config_file*}
  --InputFile={*input_file*}
  --InputBitDepth=10
  --OutputBitDepth=10
  --OutputBitDepthC=10
  --InputChromaFormat=444
  --FrameRate={*frame_rate*}
  --DecodingRefreshType=2
  --FramesToBeEncoded=97
  --SourceWidth={*width*}
  --SourceHeight={*height*}
  --IntraPeriod=32
  --QP={*qp*}
  --Level=6.2
  --BitstreamFile={*bitstream_file*}

## A.4 Visual Quality Comparison

The visual quality comparison is shown in Figure 12. It can be seen that our method has better reconstruction of detailed textures with lower bitrates, especially the parts accompanied by intense motion and obscuration.

## A.5 Effectiveness of multi-scale optical flows

The optical flows at each scale contribute to a noticeable performance improvement. To investigate the role at each scale, we designed the following experiment: we iteratively replace the optical flows at one scale with the downsampled/upsampled version from another scale and then fine-tune the network until convergence. The results are in Table 4, where "$(\cdot)$ ↑$_2$" denotes 2× upsample, "$(\cdot)$ ↓$_2$" denotes 2× downsample. "$v^1, v^2, v^3, \hat{v}^1, \hat{v}^2, \hat{v}^3$" are all shorthand for the corresponding scales: specifically, "$v^1$" denotes $v_{t,f}^1, v_{t,b}^1$, and likewise for the others. For example, Method M1 replaces the largest-size flows $v^1, \hat{v}^1$ with the intermediate-scale flows $v^2, \hat{v}^2$, its result measures the

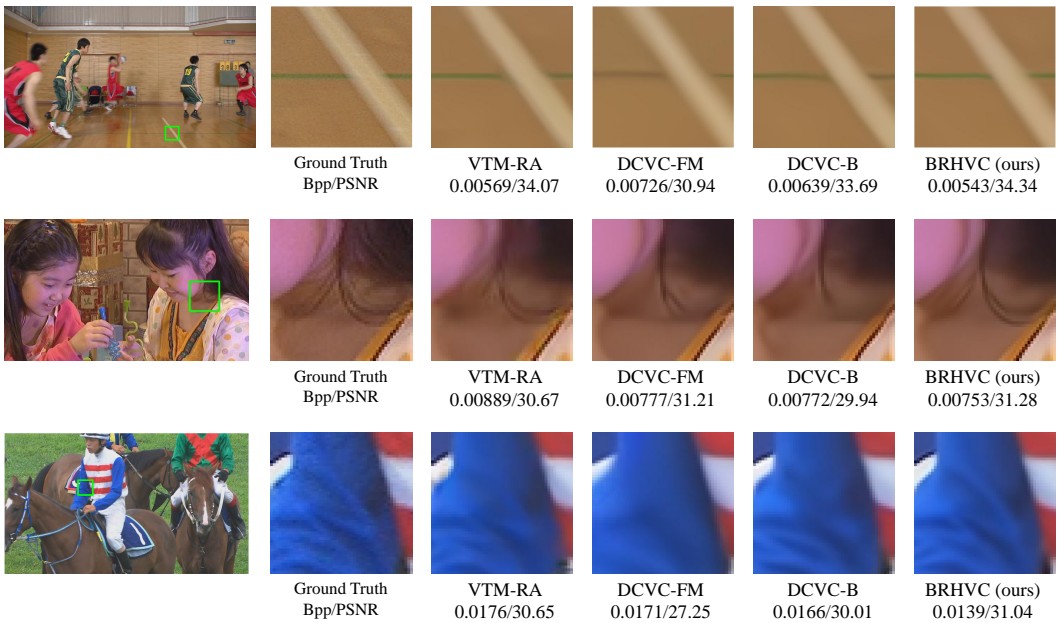

Figure 12: Visual quality comparison.

Table 4: Contributions of the optical flows at different scales.

| Method | Encoder Flows | Decoder Flows | BD-rate |
|--------|---------------|---------------|---------|
| BRHVC | – | – | 0 |
| M1 | $v^1 \leftarrow (v^2) \uparrow_2$ | $\hat{v}^1 \leftarrow (\hat{v}^2) \uparrow_2$ | 6.7% |
| M2 | $v^2 \leftarrow (v^1) \downarrow_2$ | $\hat{v}^2 \leftarrow (\hat{v}^1) \downarrow_2$ | 4.2% |
| M3 | $v^3 \leftarrow (v^2) \downarrow_2$ | $\hat{v}^3 \leftarrow (\hat{v}^2) \downarrow_2$ | 3.0% |

performance change caused by the absence of $v^1, \hat{v}^1$. Likewise, M2 and M3 denote replacing the intermediate- and smallest-size optical flows, respectively.

The results show that the flow at every scale provides a noticeable gain, demonstrating that leveraging additional flows indeed enables more effective motion compensation. Moreover, optical flows at different scales exhibit varying contributions. As the spatial resolution decreases, the contribution of the optical flow at that scale diminishes due to the reduced amount of information it carries.

### A.6 Visualization of BMC

As shown in Figure 13, the camera locks onto the moving racehorse, and the motion of the background should be consistent. Due to the limited receptive field of the flow generation network, it is prone to losing track of the moving target when the span is large, resulting in disordered estimated motion in the background (e.g., $v_{4,0}^1$ in BRHVC and $v_{4,0}$ in the baseline). However, BMC can enhance compressed flows (e.g., $\hat{v}_{4,0}^1$) with large-scale flows (e.g., $v_{4,0}^2$ and $v_{4,0}^3$), thereby achieving more accurate motion compensation for large spans.

### A.7 Model Comparison

**Details of our baseline.** At the onset of this work, given that DCVC-B had not yet released its source code, we referred to its design to reimplement the Baseline. Subsequently, once the DCVC-B framework became open-source, we re-trained it using our training settings to obtain DCVC-B (re-trained). As a result, there are differences in the implementation details between the two. The main difference is as follows:

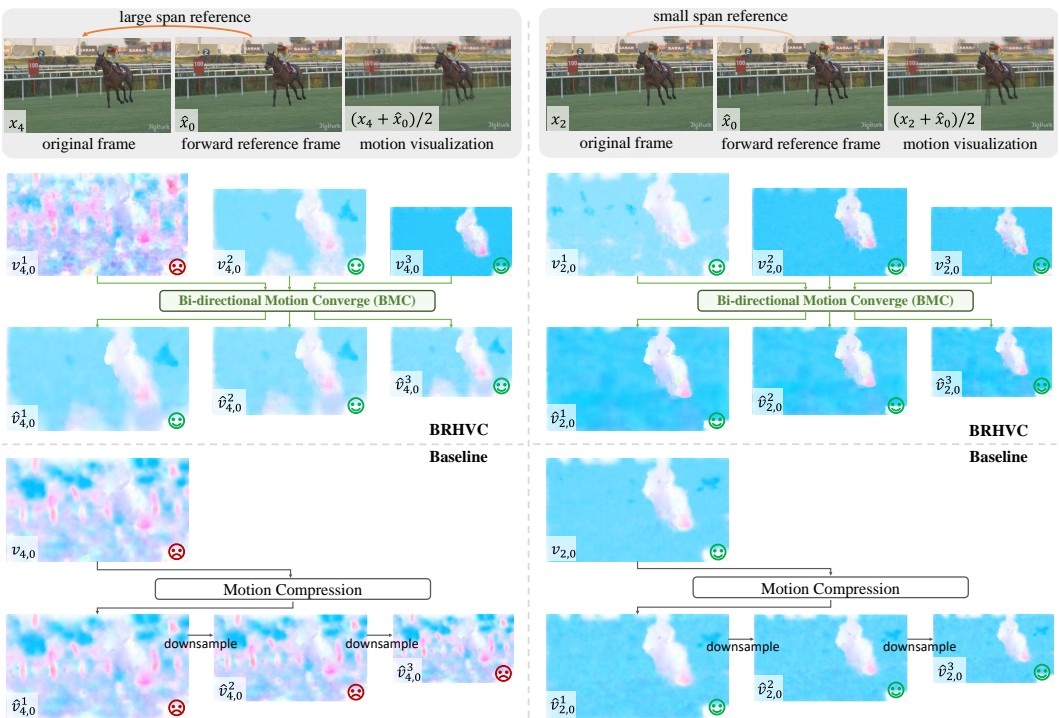

Figure 13: Comparison of motion compression between BRHVC and the baseline. We only present the forward motion and omit the backward motion. The smiling face denotes regular and consistent background motion, while the sad face denotes irregular and disordered background motion.

The Baseline and BRHVC first dimensionize the references to match the dimensions of latent features and then feed them into Encoder/Decoder, while DCVC-B directly inputs them. Taking the largest size as an example, the process of integrating reference information into the Encoder is as follows:

DCVC-B: $y_r^1 = Conv^1(cat[x, C_f^1, C_b^1])$,

Baseline: $C_t^1 = RB^1(C_f^1, C_b^1)$ then $y_r^1 = Conv^1(cat[x, C_t^1])$,

BRHVC: $C_t^1, ... = BCF_E(C_f^1, C_b^1, ...)$ then $y_r^1 = Conv^1(cat[x, C_t^1])$,

where $y_r^1$ denotes the latent feature integrated with reference information, $Conv^1$ denotes a convolution layer in the Encoder, $cat[\cdot]$ denotes concatenation along the channel dimension, $RB^1$ denotes stacking of some ResBlocks. It should be noted that the tensor $C_t$ has the same dimensions as $C_f$ and $C_b$, which reduces the computational overhead of the Baseline and BRHVC in the Encoder/Decoder compared to DCVC-B. As a result, the Baseline has fewer kMACs and memory overhead compared to DCVC-B, but its decoding speed is slightly slower. This may indicate that there is still room for optimization in our code, which we will attempt to improve in future work. Moreover, since there is not much difference in the overall mechanism, the performance of the two is also quite close.

Table 5: Comparison with models of the comparable complexity or encoding/decoding speed.

| Model | BD-rate | Parameters | kMACs/pixel | Memory | Enc. / Dec. time |
|---|---|---|---|---|---|
| DCVC-B(re-trained) | 0.4% | 24.4 M | 2934.9 | 13.5 G | 646 / 511 ms |
| Baseline (BL) | 0.0 | 23.7 M | 2831.9 | 9.6 G | 681 / 538 ms |
| BL-1 | -3.9% | 28.4 M | 4060.9 | 10.4 G | 879 / 670 ms |
| BL-2 | -4.1% | 31.7 M | 4833.9 | 11.1 G | 983 / 671 ms |
| BRHVC | -12.3% | 29.4 M | 3887.8 | 12.4 G | 983 / 670 ms |

**Comparison with additional models.** We construct two additional models for comparison to verify whether simply stacking modules can yield comparable gains. To match the complexity of BRHVC,

*BL-1* stack 4 ResBlocks per scale (three scales in total) for reference fusion in the encoder and decoder. To match the encoding/decoding speed of BRHVC, *BL-2* stack 9 and 4 ResBlocks in the encoder and decoder, respectively. In contrast, the baseline employs only 1 ResBlock per scale at both the encoder and decoder.

Table 5 shows that BL-1 can save 3.9% and 4.1% bitrates than our baseline, far below BRHVC's 12.3% bitrate saving with a relatively small GPU memory cost. This indicates that simply stacking network modules can hardly further improve performance, while the task-specific network design of BRHVC plays a significant role.

## A.8 MS-SSIM performance

We adopt MS-SSIM, the most commonly used alternative metric besides PSNR, for evaluation. Following the DCVC-B setup, we fine-tune our MS-SSIM model for only 2 epochs, starting from the MSE-optimized model. The results are shown in Table 6.

Table 6: BD-rate comparison for MS-SSIM. DCVC-B and BRHVC are both optimized by MS-SSIM in fine-tuning.

| Method | HEVC B | HEVC C | HEVC D | HEVC E | Average |
|---|---|---|---|---|---|
| VTM-LD | 0 | 0 | 0 | 0 | 0 |
| VTM-RA | -35.70% | -30.30% | -31.51% | -30.53% | -31.0% |
| DCVC-B | -43.87% | -43.18% | -58.50% | -47.78% | -42.4% |
| BRHVC (ours) | -45.58% | -45.48% | -59.97% | -45.83% | -44.0% |

## A.9 Experiments with different Intra Periods

We conducted experiments with different IP settings in Table 7. Since the configuration files provided by HM-RA and VTM-RA have a maximum IP of 32, we performed experiments with IP settings of 8, 16, and 32. For the sake of expediency, we use the first 33 frames on HEVC for testing. Moreover, since the bitrate proportion of I-frames varies significantly across different IP settings, we also provide the BD-rate computed only over B-frames, denoted as BD-rate w/o I-frame.

Table 7: BD-rate variations across different IP settings on the first 33 frames.

| Method | Anchor | BD-rate | BD-rate w/o I-frame |
|---|---|---|---|
| BRHVC (IP8) | DCVC-B (IP8) | -2.0% | -15.3% |
| BRHVC (IP16) | DCVC-B (IP16) | -6.2% | -17.8% |
| BRHVC (IP32) | DCVC-B (IP32) | -8.8% | -21.6% |

The results show that, as IP increases, the frame span also increases in some hierarchical levels, thereby increasing the severity of the URC problem. Under this condition, the improvement of BRHVC over DCVC-B also increases (from -15.3% to -21.6%). This indicates that our method successfully handles URC while DCVC-B does not, thereby aligning with our motivation.

