# OpenReview forum: "Neural B-frame Video Compression with Bi-directional Reference Harmonization"
_NeurIPS.cc/2025/Conference — NeurIPS 2025 poster_

### Official Review · Reviewer_L376 · 2025-06-28

**Clarity:** 4
**Significance:** 4
**Originality:** 4
**Rating:** 5
**Confidence:** 5

**Summary:**

The authors analyse the effect of the Unbalanced Reference Contribution (URC) issue with a quantitative experiment in Section 3. Based on the motivation of coping with URC issue, they build a novel framework termed BRHVC, with proposed BMC and BCF. Experimental results show that their BRHVC outperforms previous SOTA nerual video compression methods, which is a remarkable progress.

**Questions:**

See Weaknesses.

**Ethical Concerns:**

["NO or VERY MINOR ethics concerns only"]

**Final Justification:**

Taking into account the paper's contributions and the additional evidence provided during the rebuttal, I have decided to raise my score to 5. The authors identify and analyze the URC issue in bi-directional compression and, building on this insight, propose BRHVC, which substantially outperforms previous Neural Video Compression (NVC) methods. In the rebuttal, they supplied further ablation studies and complexity comparisons, demonstrating that the design is well-motivated and the computational overhead is acceptable.

**Limitations:**

As mentioned in paper, BRHVC does not perform sufficiently well on high frame rate datasets, which may require further investigation in future work.

**Quality:**

4

**Strengths And Weaknesses:**

Strengths:
1. The proposition and analysis of the URC issue are very interesting. The authors design a clever toy experiment to illustrate the significant impact of URC on bi-directional compression.
2. The motivation behind the design of BRHVC is reasonable. The BMC can provide more accurate optical flows in a large scale, while the BCF offers reference weights to better integrate reference information.
3. The progress of BRHVC is remarkable compared with the baseline. And the overall performance outperforms previous SOTA nerual video compression methods, which is comparable to VTM-RA.
4. The paper is well-written with good readability and is easy to understand.

Weaknesses:
1. The description of the baseline is too brief. It seems that the baseline employs a similar architecture to DCVC-B [33], and its performance is also quite comparable. So why not directly use DCVC-B as the baseline? What are the main differences between the baseline and DCVC-B?
2. There is a lack of ablation study. Although, as mentioned above, the improvement of BRHVC over the baseline is reasonable, there is still a need for a model with similar complexity to be compared with BRHVC.

---

> ### Author Rebuttal · Authors · 2025-07-31
>
> We sincerely appreciate your affirmation and recognition of our work. Here are our detailed responses to your concerns, and we hope they will clarify our work.
>
> **Weakness 1: The difference between Baseline and DCVC-B (re-trained).**
>
> At the onset of this work, given that DCVC-B had not yet released its source code, we referred to its design to reimplement the *Baseline*. Subsequently, once the DCVC-B framework became open-source, we re-trained it using our training settings to obtain *DCVC-B (re-trained)*. As a result, there are differences in the implementation details between the two. The main difference is as follows:
>
> The Baseline and BRHVC first dimensionizes the references to match the dimensions of latent features and then feeds them into Encoder/Decoder, while DCVC-B directly input them. Taking the largest size as an example, the process of integrating reference information into the Encoder is as follows:
>
> DCVC-B: $y^1_r=Conv^1(cat[x,C^1_f,C^1_b])$,
>
> Baseline: $C^1_t=RB^1(C^1_f, C^1_b)$ then $y^1_r=Conv^1(cat[x,C^1_t])$,
>
> BRHVC: $C^1_t,...=BCF_E(C^1_f, C^1_b,...)$ then $y^1_r=Conv^1(cat[x,C^1_t])$,
>
> where $y^1_r$ denotes the latent feature integrated with reference information, $Conv^1$ denotes a convolution layer in the Encoder, $cat[\cdot]$ denotes concatenation along the channel dimension, $RB^1$ denotes stacking of some ResBlocks.
>
> It should be noted that tensor $C_t$ has the same dimensions as $C_f$ and $C_b$, which reduces the computational overhead of the Baseline and BRHVC in the Encoder/Decoder compared to DCVC-B.
>
> As a result, the Baseline has fewer kMACs and memory overhead compared to DCVC-B, but its decoding speed is slightly slower. This may indicate that there is still room for optimization in our code, which we will attempt to improve in future work. Moreover, since there is not much difference in the overall mechanism, the performance of the two is also quite close.
>
> **Weakness 2: Comparison under equivalent complexity.**
>
> This question is similar to Reviewer BaaM's Question 1. As shown in Table 4, we constructed a more complex model (denoted as *Baseline w/ Resblocks*) for comparison. Experimental results show that merely increasing model complexity to match BRHVC’s encoding/decoding speed yields only a 4.1 % improvement—far below BRHVC’s 12.3 %. It confirms that BRHVC indeed enhances the performance effectively.

---

> > ### Comment · Reviewer_L376 · 2025-08-03
> > **Further Comment**
> >
> > Thank the authors for the detailed responses.
> >
> > The other reviewers (Reviewer iau7 and BaaM) raised several valuable questions, and the authors have provided comprehensive answers. After reviewing their replies, I have noted several important issues beyond complexity that I had previously overlooked, such as the URC effect, optical flow concerns and dataset limitations. However, I still have some further questions below.
> >
> > 1) Could you provide some cases to demonstrate how BRHVC handles the URC problem? For instance, does BRHVC perform better on video sequences with complex motion, such as RaceHorses or BasketballDrill?
> >
> > 2) While the authors have demonstrated the impact of resolution, could you also clarify how frame rate affects performance? In fact, the frame rate of UVG is 120 FPS, whereas that of MCL-JCV is about 24 FPS, could it affect performance?

---

> ### Author Response · Authors · 2025-08-03
>
> We are pleased to see your feedback and thank you for your timely and constructive comments. Below are our explanations in response to your supplementary questions.
>
> **Supplementary Question 1: Cases of complex motions.**
>
> The experiments in Table 10 demonstrate that **BRHVC significantly outperforms DCVC-B on datasets with more complex motions**. The sequences *BasketballDrill* and *RaceHorses* represent complex motion scenarios, *ParkScene* represents a relatively ordinary motion scenario, and *FourPeople* represents a scene with small motion. Specifically, *BasketballDrill* is a multi-person basketball scene with many moving targets and complex body movements; *RaceHorses* is a multi-person horse-riding scene with numerous occlusions; *ParkScene* is a park scene with a few people cycling by, featuring relatively stable motion; and *FourPeople* is an indoor meeting scene with small motion.
>
> **Table 10:** Performance comparison between BRHVC and DCVC-B on representative sequences in the HEVC dataset.
>
> | Sequence | Sub-dataset | DCVC-B | BRHVC |
> | --- | --- | --- | --- |
> | ParkScene | Class B | 0   | -9.9 |
> | BasketballDrill | Class C | 0   | -19.1 |
> | RaceHorses | Class D | 0   | -18.5 |
> | FourPeople | Class E | 0   | -2.3 |
>
> Experimental results show that BRHVC achieves the highest gains in the complex motion sequences *BasketballDrill* and *RaceHorses*, with improvements of 19.1% and 18.5%, respectively. In contrast, the smallest gain is observed in the small motion sequence *FourPeople*, with only 2.3%. In the ordinary scene *ParkScene*, a 9.9% improvement is achieved. Since the URC issue becomes more severe with more complex motions, these results indicate that **BRHVC is more effective at addressing the URC issue.**
>
> **Supplementary Question 2: Impact of frame rate.**
>
> Theoretically, for the same content, a lower frame rate implies larger inter-frame motion, which amplifies the impact of the URC issue. Among the 6 datasets, MCL-JCV has the lowest overall frame rate and the highest resolution, which may lead to the most severe URC issue. Table 1 shows that DCVC-B's average BD-rate is **-17.2%**, but only **-2.0%** on MCL-JCV. In contrast, BRHVC's average BD-rate is **-27.3%**, and **-16.1%** on MCL-JCV. It indicates that the network architecture designed to address the URC issue can effectively compensate for the previous shortcomings.

---

> > ### Comment · Reviewer_L376 · 2025-08-04
> > **Further comment**
> >
> > Thank you for your answer, it has resolved most of my concerns. I have one more question regarding your response to Reviewer iau7's further question.
> >
> > How can RA methods specifically implement the “parallel speed optimization”, as you mentioned?

---

> > > ### Author Response · Authors · 2025-08-04
> > >
> > > Thank you for your comments. Theoretically, RA can implement parallel speed optimization on the following parts:
> > >
> > > First, frames at each hierarchical level of RA can be encoded/decoded simultaneously. For an Intra Period of 32, RA theoretically requires only 5 inference steps to encode/decode 31 frames. Therefore, under parallel coding, the average per-frame time overhead of the added module can be reduced to as low as 5/31 of that in the single-frame case, thereby mitigating the impact of the slowdown to a certain degree.
> > >
> > > Second, certain operations involving the two reference frames can be performed simultaneously, such as feature extraction and motion compensation (see Equation 1 in the main text). In this part, the two reference frames undergo similar yet independent processing procedures. Currently, we process the two references sequentially. However, it can be parallelized in the future, which may introduce some memory overhead but would save encoding and decoding time.

---

> > > > ### Comment · Reviewer_L376 · 2025-08-05
> > > > **Further Comment**
> > > >
> > > > Thank you for the detailed responses. The clarifications and additional analyses have addressed my concerns and strengthened the paper. I will raise my rating.

---

> > > > > ### Author Response · Authors · 2025-08-05
> > > > >
> > > > > We sincerely appreciate your recognition of our efforts. Thank you very much for your time and expert advice.

---

### Official Review · Reviewer_iau7 · 2025-06-30

**Clarity:** 3
**Significance:** 2
**Originality:** 3
**Rating:** 4
**Confidence:** 3

**Summary:**

This paper introduces BRHVC, a novel neural B-frame video compression method designed to tackle the challenge of unbalanced reference contributions in hierarchical coding. It achieves this through two proposed modules: Bi-directional Motion Converge (BMC) for improved motion handling and Bi-directional Contextual Fusion (BCF) for adaptively weighting bi-directional references. BRHVC demonstrates state-of-the-art performance, outperforming previous neural methods and surpassing VTM-RA on HEVC datasets.

**Questions:**

- Could you please clarify the exact differences between the "Baseline" model used in the ablation study and the "DCVC-B (re-trained)"? What accounts for the small performance difference shown in Table 2?
- The experiments focus on an Intra Period (IP) of 32. How does BRHVC's performance, and the severity of the Unbalanced Reference Contribution problem, vary with different IP settings? Would BRHVC maintain its advantage at shorter or longer IP values?
- The performance gains are less significant on high frame rate datasets like MCL-JCV and UVG. What are the potential reasons or hypotheses for this observation? Are there specific characteristics of these datasets that limit BRHVC's effectiveness compared to HEVC datasets?

**Ethical Concerns:**

["NO or VERY MINOR ethics concerns only"]

**Final Justification:**

The paper tackles an important problem in hierarchical B-frame coding, with well-designed BMC and BCF modules delivering strong SOTA results. The rebuttal addressed most earlier concerns with additional experiments and analyses. While the computational overhead is notable, the justification is reasonable for many scenarios. Given its technical merit, clear gains, and potential for further optimization, I raise my rating to borderline accept.

**Limitations:**

Yes

**Quality:**

3

**Strengths And Weaknesses:**

** Pros
 - The paper effectively identifies and proposes solutions for the Unbalanced Reference Contribution problem inherent in hierarchical B-frame coding.
 - It achieves SOTA performance, notably surpassing the high-performance VTM-RA codec on standard datasets.
 - It introduces innovative BMC and BCF modules that enhance motion compensation and context fusion for better reference utilization.

** Cons
 - The proposed framework integrates BMC and BCF modules, where BMC refines motion estimation and compensation, and BCF uses the resulting context features for fusion. This suggests a sequential flow of information. One may consider a mechanism for tighter integration, such as joint optimization or feedback loops, where the effectiveness of the context fusion (BCF output) could inform or adapt the motion processing (BMC). Exploring a more deeply intertwined design between motion and context modules might potentially enable further performance improvements by allowing them to learn to cooperate more effectively end-to-end.

 - As shown in the ablation study (Table 2), BRHVC introduces a substantial increase in encoding time (about 50% over baseline) and decoding time (about 30% over baseline) compared to the baseline model. While the authors state this overhead is "relatively acceptable" given the BD-rate improvement, such a significant increase in computational cost could be a major barrier for real-world deployment, especially in scenarios with strict real-time constraints or limited hardware resources (like mobile devices for decoding). The trade-off between performance and complexity might not be universally "acceptable" depending on the specific application.

 - The ablation study effectively demonstrates the contribution of BCF alone and the additional contribution of adding BMC after BCF. However, it lacks the result of using BMC alone (e.g., Baseline w/ BMC). Presenting this result would provide clearer insight into the independent contribution of the BMC module and its efficiency, as well as how its interaction with BCF leads to the final performance gain.

 - The paper primarily uses PSNR-based BD-rate to demonstrate performance gains. While PSNR is a standard objective metric in video compression, it does not always correlate perfectly with perceived visual quality by humans. Metrics like SSIM, MS-SSIM, or VMAF are often considered more representative of subjective user experience. The absence of results using these alternative metrics makes it difficult to fully assess the practical benefits of BRHVC in terms of visual perception.

 - The paper acknowledges that BRHVC's performance gains are less pronounced on high frame rate datasets such as MCL-JCV and UVG compared to HEVC datasets. While identifying this limitation is valuable, the paper does not provide a detailed analysis or hypothesis as to why this performance gap exists. Is the Unbalanced Reference Contribution problem less severe in high frame rate videos due to smaller frame spans? Are there other dominant challenges in high frame rate compression that BRHVC's current design doesn't adequately address? A deeper analysis of the causes would be crucial for guiding future research on improving NBVC for high frame rate content.

- All primary experiments are conducted using an Intra Period (IP) of 32. The hierarchical coding structure of B-frames, and consequently the characteristics of reference spans and the severity of the URC problem, can change significantly with different IP lengths. While IP=32 is a common setting for RA configurations, evaluating BRHVC's performance across a range of IP values (e.g., shorter IPs like 8 or 16, or even longer ones) would provide a more comprehensive understanding of its robustness and the generalizability of its benefits across various hierarchical coding structures.

---

> ### Author Rebuttal · Authors · 2025-07-31
>
> We sincerely thank your time, effort, and professional expertise. We have carefully considered your concerns and provided detailed answers, hoping these will make our work more solid.
>
> **Weakness 1: Enhance collaboration between the two modules.**
>
> This is a great idea. Perhaps we can implement a mechanism similar to a 2-pass or coarse-to-fine approach. It would allow the two modules to interact and harmonize with each other, thereby enhancing performance. However, it might introduce some complexity. In any case, thank you for your suggestion. We will try to incorporate it in our future work.
>
> **Weakness 2: Is the computational cost "acceptable"?**
>
> We believe the overhead is **indeed "acceptable"** given the performance gains achieved. To verify it, we use the most critical metrics—percentage overhead of decoding time over the baseline and BD-rate gain based on its baseline as the criterion.
>
> **Table 6:** Gain–overhead comparison across some methods.
>
> |     | Venue | Baseline | Gain | Overhead |
> | --- | --- | --- | --- | --- |
> | DCVC-DC | CVPR'23 | DCVC-HEM | 24.9% | 17% |
> | [s1] | CVPR'25 | DCVC-DC | 8.3% | 32% |
> | [s2] | CVPR'25 | DCVC-DC | 5.3% | 45% |
> | BRHVC (ours) | --  | DCVC-B | 10.1% | 31% |
>
> **References**
>
> [s1] Augmented deep contexts for spatially embedded video coding.
>
> [s2] Neural video compression with context modulation.
>
> The results are shown in Table 6. Although the anchor of gains may be not strictly aligned, it still provides partial evidence. It can be seen that our method’s trade-off between gain and overhead falls within an acceptable range. Several additional factors should be considered:
>
> First, coding efficiency exhibits diminishing returns: DCVC-DC (2023) achieves an attractive gain-overhead ratio, whereas the two 2025 works [s1,s2] require substantially higher costs for further improvements.
>
> Second, RA coding inherently processes more information than LD coding, so any enhancement built upon it is more prone to increased overhead. However, unlike LD coding, RA theoretically allows parallel encoding of multiple frames at the same level, which can further reduce the overall complexity when deployed in practice.
>
> Third, RA is not designed for low-latency requirements. Much like the historical development of LD, it may be more appropriate for RA to first explore performance-boosting mechanisms and then seek complexity reduction.
>
> In conclusion, we believe the overhead is indeed "acceptable" given the performance gains achieved.
>
> **Weakness 3: Performance of Baseline w/ BMC.**
>
> As shown in Table 4, using BMC alone (denoted as *Baseline w/ BMC*) can save 6.4% bitrate. Moreover, the complexity of each module has been explicitly listed.
>
> **Weakness 4 : Other image-quality metrics.**
>
> We adopt MS-SSIM, the most commonly used alternative metric besides PSNR, for evaluation. Following the DCVC-B setup, we fine-tune our MS-SSIM model for only 2 epochs, starting from the MSE-optimized model. **The results in Table 7 show that BRHVC significantly outperforms VTM-RA in terms of MS-SSIM.** Due to the time constraints of the rebuttal period, we can not thoroughly fine-tune the model, which may limit further performance gains relative to DCVC-B.
>
> **Table 7:** BD-rate comparison for MS-SSIM. DCVC-B and BRHVC are both optimized by MS-SSIM in fine-tuning.
>
> | Method | HEVC B | HEVC C | HEVC D | HEVC E | MCL-JCV | UVG | Average |
> | --- | --- | --- | --- | --- | --- | --- | --- |
> | VTM-LD | 0   | 0   | 0   | 0   | 0   | 0   | 0   |
> | VTM-RA | -35.70% | -30.30% | -31.51% | -30.53% | -28.62% | -29.59% | -31.0% |
> | DCVC-B | -43.87% | -43.18% | -58.50% | -47.78% | -37.35% | -23.91% | -42.4% |
> | BRHVC (ours) | -45.58% | -45.48% | -59.97% | -45.83% | -39.32% | -27.38% | -44.0% |
>
> **Weakness 5, Quesion 3: Performance on UVG and MCL-JCV.**
>
> This Weakness is similar to Reviewer 5ESm's Question 2. The primary reason is the absence of high-resolution training data in our dataset. Please see it for more details.
>
> **Weakness 6, Question 2: Experiments with different IPs.**
>
> We conducted experiments with different IP settings in Table 8. Since the configuration files provided by HM-RA and VTM-RA have a maximum IP of 32, we performed experiments with IP settings of 8, 16, and 32. For the sake of expediency, we use the first 33 frames on HEVC for testing. Moreover, since the bitrate proportion of I-frames varies significantly across different IP settings, we also provide the BD-rate computed only over B-frames, denoted as *BD-rate w/o I-frame*.
>
> **Table 8:** BD-rate variations across different IP settings on the first 33 frames.
>
> | Method | Anchor | BD-rate | BD-rate w/o I-frame |
> | --- | --- | --- | --- |
> | BRHVC (IP8) | DCVC-B (IP8) | -2.0% | -15.3% |
> | BRHVC (IP16) | DCVC-B (IP16) | -6.2% | -17.8% |
> | BRHVC (IP32) | DCVC-B (IP32) | -8.8% | -21.6% |
>
> The results show that, as IP increases, the frame span also increases in some hierarchical levels, thereby increasing the severity of the URC problem. Under this condition, the improvement of BRHVC over DCVC-B also increases (from -15.3% to -21.6%). This indicates that our method successfully handles URC while DCVC-B does not, thereby aligning with our motivation.
>
> **Question 1: The difference between Baseline and DCVC-B (re-trained).**
>
> This Question is similar to Reviewer L376's Weakness 1. Please see it for more details.

---

> > ### Comment · Reviewer_iau7 · 2025-08-04
> > **Further question**
> >
> > Thank you for providing answers to the questions and comments.
> > Regarding the Weakness 2, could you please provide the gain–overhead comparison on the 'encoding time' as well across other methods?

---

> > > ### Author Response · Authors · 2025-08-04
> > >
> > > We are pleased to see your response. The results incorporating encoding time are shown below. As we mentioned in our response to Weakness 1 of Reviewer L376, our baseline is not perfectly identical to DCVC-B. Therefore, we have also included the comparison results with our baseline. We define the *gain-overhead ratio* is $Gain/Overhead$, which represents the relative trade-off between gain and overhead, where a higher value indicates better performance.
> > >
> > > **Table 11:** Gain–overhead comparison across some methods. The values in "( )" represent the *gain-overhead ratio*, where higher values are better.
> > >
> > > |     | Venue | Baseline | Gain | Encoding Overhead | Decoding Overhead |
> > > | --- | --- | --- | --- | --- | --- |
> > > | DCVC-DC | CVPR'23 | DCVC-HEM | 24.9% | 13% (1.91) | 17% (1.5) |
> > > | [s1] | CVPR'25 | DCVC-DC | 8.3% | 17% (0.48) | 32% (0.26) |
> > > | [s2] | CVPR'25 | DCVC-DC | 5.3% | 41% (0.13) | 45% (0.12) |
> > > | BRHVC (ours) | --  | DCVC-B | 10.1% | 52% (0.20) | 31% (0.33) |
> > > | BRHVC (ours) | --  | Our Baseline | 12.3% | 44% (0.28) | 24% (0.51) |
> > >
> > > Table 11 shows that the encoding overhead of BRHVC is 52% relative to DCVC-B and 44% relative to our baseline. BRHVC's *encoding gain-overhead ratio*s (0.20 and 0.28) are higher than [s2] but lower than [s1]. However, BRHVC's *decoding gain-overhead ratio*s (0.33 and 0.51) are higher than both [s1] and [s2]. We believe that these overheads of BRHVC are acceptable relative to the gains. Please allow us to explain the reasons:
> > >
> > > First, in most compression scenarios, the decoding side typically has weaker computational power, so the decoding complexity is more crucial in determining the deployment and application of the entire algorithm. Additionally, our *decoding gain-overhead ratios* are superior to [s1][s2], which can be more beneficial for practical applications.
> > >
> > > Second, our BRHVC trades off some speed to save memory. As shown in Table 4, BRHVC has an even **smaller memory overhead (12.4G)** compared to DCVC-B **(13.5G)**, thereby facilitating deployment on a wider range of devices.
> > >
> > > Third, as previously mentioned, RA methods have more potential for parallel speed optimization and lower requirements for coding latency compared to LD methods. Therefore, our BRHVC prioritizes performance over latency.
> > >
> > > In summary, we believe that these overheads are acceptable compared to our performance gains. However, we are also aware of the necessity to reduce complexity and will strive to achieve it in our future work. Finally, thank you once again for your response.

---

> > > ### Author Response · Authors · 2025-08-08
> > >
> > > Thank you once again for your constructive comments. In our previous response, we conducted experiments using the MS-SSIM metric and provided both experiments and analysis to explain the results obtained on the UVG and MCL-JCV datasets. In addition, we included evaluations under various IP settings to demonstrate that our method remains aligned with the original motivation about URC.
> > >
> > > Furthermore, we provided a detailed justification for why we believe BRHVC’s complexity is acceptable, as you specifically raised this concern.
> > >
> > > We would greatly appreciate it if you could confirm whether these clarifications fully address your concerns. As the discussion phase draws to a close, we sincerely look forward to your further feedback.

---

> > > > ### Comment · Reviewer_iau7 · 2025-08-09
> > > >
> > > > Thank you for the additional clarifications and the follow-up results.
> > > > Most of my earlier concerns such as the absence of BMC-only results, the lack of MS-SSIM evaluation, and missing analysis for different IP values have been well addressed, and the explanations for reduced gains on high-frame-rate datasets are reasonable. While the computational overhead is still notable, the gains are meaningful and decoding efficiency compares favorably to recent works.
> > > > Combined with the technical merit and potential for further optimization, these points lead me to raise my rating to borderline accept.

---

> > > > > ### Author Response · Authors · 2025-08-09
> > > > >
> > > > > We sincerely thank you for your positive feedback and the increased rating. We are grateful for the time and expertise you have devoted to reviewing our paper.

---

### Official Review · Reviewer_5ESm · 2025-07-01

**Clarity:** 3
**Significance:** 2
**Originality:** 3
**Rating:** 4
**Confidence:** 3

**Summary:**

This paper proposes BRHVC, a novel neural B-frame video compression method that addresses the inaccurate motion compensation and unbalanced reference contribution issue in bi-directional coding. The paper introduces two key modules(BMC and BCF) to enhance motion compensation and adaptively harmonizes reference information. Despite underperforming VTM-RA on several test datasets, the paper provides visualized analysis of module design and effect, and includes runtime efficiency evaluation, offering a well-rounded and transparent study.

**Questions:**

1. The paper states that the Motion Compensation module adopts the Contextual-Feature Extraction (CFE) method as used in references [20] and [33]. However, this specific concept does not appear explicitly in the main texts of those references. Could the authors clarify what is meant by this term?

2. Could the authors provide a theoretical analysis of why the proposed system performs less effectively on high-resolution datasets, such as UVG and MCL-JCV?

3. How is the proposed method compared to recent multimodal-based video compression methods, such as M3-CVC?

**Ethical Concerns:**

["NO or VERY MINOR ethics concerns only"]

**Final Justification:**

Thanks for the authors' feedback, which mainly addresses my concerns. I will keep my score.

**Limitations:**

yes

**Quality:**

3

**Strengths And Weaknesses:**

Strength:

1. Multiscale optical flow extraction is employed in this paper, which is a novel approach that effectively mitigates inaccuracies in motion estimation. Its effectiveness is further supported by clear visualizations provided in the appendix.

2. The BCF module is well-designed to leverage multiscale motion-compensated features through a weighted summation scheme to improve contextual reconstruction. The approach demonstrates a certain level of originality.

3. The authors perform a quantitative evaluation of runtime efficiency, including encoding and decoding time, which contributes to assessing the system’s practical applicability.


Weakness:

1. The multiscale optical flow extraction process effectively requires three separate calls to SpyNet for each frame, which may incur redundant computation. As shown in Figure 13 of the appendix, when the motion between frames is small, extracting optical flow at the original resolution and directly downsample can yield comparable results, suggesting the current approach may be computationally excessive in such cases.

2. The paper lacks detailed reporting on the overall computational complexity (e.g., in terms of kMACs/pixel), as well as comparisons with similar systems. In addition, it does not provide a breakdown of the computational cost and latency associated with each individual module.

3. The proposed system underperforms the VTM-RA baseline on 4 out of 6 benchmark datasets, indicating that its overall compression performance still requires further improvement.

---

> ### Author Rebuttal · Authors · 2025-07-31
>
> We sincerely thank you for your positive feedback on our work. Below are our responses to your concerns, and we hope these efforts can better clarify our work.
>
> **Weakness 1: Computationally excessive of BMC.**
>
> It is indeed the case that invoking SpyNet three times may seem excessive, especially for small motions, as you pointed out. However, as shown in Reviewer BaaM's Question 2, on average, the optical flow at each scale has made an indispensable contribution. Thus, we think retaining three separate flows remains the best compromise for now. In future work, we intend to migrate to a single-pass architecture that delivers all three flows in one inference. Thank you for your comment, it has reinforced this plan. Nevertheless, BRHVC currently still achieves good performance under the same complexity (see *Baseline w/ Resblocks* in Table 4).
>
> **Weakness 2: More detailed computational complexity.**
>
> Refer to Table 4, we provide more detailed metrics, including kMACs/pixel and max memory usage. Specifically, BCF adds approximately 700 kMACs/pixel and 50 ms to decoding latency, while BMC adds about 340 kMACs/pixel and 80 ms. Although BRHVC increases complexity, as can be seen from Table 4, our method is more effective for methods of equivalent complexity (i.e., *Baseline w/ Resblocks*).
>
> **Weakness 3: Performance comparison with VTM-RA.**
>
> Although BRHVC underperforms VTM-RA on some datasets, it indeed achieves a substantial improvement over previous Nerual Video Compression (NVC) methods, outperforming them on all datasets (see Table 1). Additionally, the gap with VTM-RA is mainly observed on UVG and MCL-JCV, which will be analyzed in Question 2.
>
> **Question 1: More details on Contextual-Feature Extraction (CFE).**
>
> The CFE module combines Temporal Context Mining [20, 32] with the post-processing Group-Based Offset Diversity mechanism [20]. For conciseness, we merge the two modules into one, denoted as CFE. The role of CFE is to extract reference information from $F_f,F_b$ and perform motion compensation on them with optical flows. To convey this point more clearly, we will refine the wording in this part. Thank you for your feedback.
>
> **Question 2: Performance on UVG and MCL-JCV.**
>
> We speculate that the main reason for our inferior performance compared to VTM-RA on MCL-JCV and UVG is that these datasets are high-resolution (1920×1080), whereas the Vimeo-90K training set commonly used for Neural Video Compression (NVC) lacks such high-resolution data. Besides BRHVC, this shortcoming results in generally poor performance of NVCs methods on these datasets.
>
> To verify this, we downsampled the two datasets and re-evaluated them against VTM-RA. We resized the images to 480×270 and, for efficiency, selected the first 33 frames for testing. As shown in Table 9, **the performances of NVC methods overall show significant improvements in low-resolution settings, with BRHVC even outperforming VTM-RA noticeably.**
>
> **Table 9:** BD-rate comparison on downsampled MCL-JCV and UVG.
>
> | Method | MCL-JCV (downsample) | UVG (downsample) |
> | --- | --- | --- |
> | HM-RA | 35.4% | 42.5% |
> | VTM-RA | 0   | 0   |
> | DCVC-DC | 5.6% | 6.8% |
> | DCVC-B | 13.9% | 0.9% |
> | BRHVC | -5.3% | -10.5% |
>
> Most NVC methods train on cropped 256×256 patches, which aids training stability and conserves computational resources but limits the model's ability to handle larger spatial contexts (e.g., 1080P images). We encourage the NVC community to further explore this issue, which may lead to further improvements in the performance of NVC. Despite these challenges, our method has made considerable progress, bringing NVC closer to VTM-RA than ever before.
>
> **Question 3: Comparison with multimodal-based method.**
>
> Regrettably, we believe it may not be feasible to make a fully fair comparison between BRHVC and these approaches. First, there are differences in research focus: their methods target subjective metrics (e.g., LPIPS) at extremely low bitrate, while our work aims for objective metrics (i.e., PSNR) and video fidelity at medium to high bitrate. Second, the lack of open-source code and PSNR results for M3-CVC further complicates direct comparison. Nevertheless, we will open-source our model in the future, enabling other researchers to compare it with other video coding methods across different fields.

---

> > ### Comment · Reviewer_5ESm · 2025-08-04
> > **response**
> >
> > Thanks for the authors' feedback, which mainly addresses my concerns. I will keep my score.

---

> > > ### Author Response · Authors · 2025-08-04
> > >
> > > Thank you for your valuable feedback and for recognizing our work. We truly appreciate your support.

---

### Official Review · Reviewer_BaaM · 2025-07-02

**Clarity:** 3
**Significance:** 2
**Originality:** 3
**Rating:** 4
**Confidence:** 4

**Summary:**

In this paper, the authors propose a new video coding model, Bi-directional Reference Harmonization Video Compression (BRHVC). The model is based on the conditional coding framework; Unlike most existing work, which focuses on low-delay settings, BRHVC targets hierarchical B-frame coding. While existing learned B-frame coding models offer limited gains over low-delay methods, the authors tackle these limitations by (1) utilising multi-scale optical flow to warp features at different scales and jointly encoding these flows into the latent representation, which improves performance on sequences with large motions, and (2) explicitly modelling the weighting of different reference contexts, enabling better use of bi-directional information.

**Questions:**

- How exactly were the experiments performed when removing the BCF/BMC modules?
- The authors should also validate the use of different motion vectors at each scale.
- It may help to clarify the two main B-frame coding strategies - low-delay B and hierarchical B. The proposed model only address the latter category.

**Ethical Concerns:**

["NO or VERY MINOR ethics concerns only"]

**Final Justification:**

In this paper, the authors propose a new B-frame coding model that exploits both past and future reference frames, especially when those references contribute unequally. Experiments show consistent performance gains over prior methods.

However, while the paper introduces the notion of “unbalanced reference contribution” as its key motivation, it does not demonstrate how existing models are limited by this phenomenon. In principle, with sufficient data and capacity, end-to-end optimised compression models could learn to balance reference contributions, so handcrafting modules for handling this phenomenon does not seem to be necessary; showing that they fail to do so would substantially strengthen the work’s insight.

But overall, the authors have addressed most of my other concerns, including additional experiments that support the design, thus I have adjusted my score accordingly.

**Limitations:**

The authors have discussed some limitations of the proposed methods. Please refer to the weaknesses listed above.

**Quality:**

2

**Strengths And Weaknesses:**

Overall, although the proposed model demonstrates improvement over the baselines, its novelty is modest, and the new modules have not yet been validated comprehensively.

Strengths
- The proposed model focuses on hierarchical B-frame coding, addressing a major limitation of current neural video codecs.
- The motivation is sound: balancing the use of multiple references is indeed important for video coding.

Weaknesses
- While the proposed techniques are verified by an ablation study showing BD-rate gains for each component, the experiments are not comprehensive. In particular, adding these modules increases both computational complexity and model size. It is unclear whether the added components are the best choice - could similar gains be achieved by simply stacking more layers at the same positions with comparable complexity?
- The memory footprint (a limitation of conditional coding models) and complexity (e.g., MACs per pixel) are not provided. While encoding/decoding speeds are reported, memory usage and MACs are also critical for practical application, since the decoding speed can vary significantly across platforms.
- Although the experiments show that there are different importance of references, the authors did not demonstrate how existing models fail to leverage bi-directional context. This omission weakens the motivation for the paper.

---

> ### Author Rebuttal · Authors · 2025-07-31
>
> We sincerely thank you for your detailed and instructive reviews, and for recognizing our contributions to B-frame coding. Below are some responses that we hope will address your concerns.
>
> **Weakness 1,2: Additional metrics and comparison under equivalent complexity.**
>
> We conducted more comprehensive comparison on additional metrics such as kMACs and Max Memory. Furthermore, to verify whether simply stacking modules can yield sufficient gains, we construct a more complex model (denoted as *Baseline w/ Resblocks*) for comparison. The results are as follows:
>
> **Table 4:** Ablation study on different network designs (based on Table 2 in the main text).
>
> | Method | BD-rate | Parameters | kMACs/pixel | Max Memory | Encoding time | Decoding time |
> | --- | --- | --- | --- | --- | --- | --- |
> | DCVC-B (re-trained) | 0.4% | 24.4M | 2934.9 | 13.5G | 646 ms | 511 ms |
> | Baseline | 0   | 23.7M | 2831.9 | 9.6G | 681 ms | 538 ms |
> | Baseline w/ BMC | -6.4% | 24.3M | 3170.9 | 10.6G | 780 ms | 602ms |
> | Baseline w/ BCF | -6.6% | 28.9M | 3532.8 | 11.4G | 896 ms | 583 ms |
> | Baseline w/ BCF,BMC (BRHVC) | -12.3% | 29.4M | 3887.8 | 12.4G | 983 ms | 670 ms |
> | Baseline w/ Resblocks | -4.1% | 31.7M | 4833.9 | 11.1G | 983 ms | 671 ms |
>
> Considering that inference speed is influenced by many factors and may not always correlate directly with kMACs or memory usage, as noted in [15] and other studies, we suggest using actual coding time as a more reliable metric in this case. Experimental results show that merely increasing model complexity to match BRHVC’s encoding/decoding speed yields only a **4.1 %** improvement—far below BRHVC’s **12.3 %**. It indicates that **the improvement of BRHVC stems from the motivation behind our design, rather than merely from increased complexity.** Moreover, Table 6 in Reviewer iau7's Weakness 2 indicates that **the increase in complexity resulting from our method design is acceptable.**
>
> Here is a detailed description of *Baseline w/ Resblocks*. To match the encoding/decoding speed of BRHVC, we stack 9 ResBlocks per scale (three scales in total) for reference fusion in the encoder and 4 ResBlocks per scale in the decoder. In contrast, the baseline employs only 1 ResBlock per scale at both the encoder and decoder. Please refer to Reviewer L376’s Weakness 1 for additional details of the Resblocks (denoted as $RB$).
>
> Further ablation experiments on BMC can be found in Question 2.
>
> **Weekness 3: How existing models fail to use reference information.**
>
> In the main text, we have provided explanations of our design principles based on existing models and demonstrated the visualization effects. To further validate the effectiveness, we conducted additional comparative experiments. Below are the details.
>
> First, existing methods did not have specialized designs for the URC issue. As stated in lines 170–174, existing methods [33,34] do not explicitly model the weights of the two reference frames; instead, they perform reference fusion directly via concatenate&convolution. The weights of references are learned in training and fixed after training, which can not adapt to the value of references. In contrast, as shown in Equation 3, our BRHVC explicitly models the weights with $s_f$, $s_b$, and $b_t$.
>
> Second, the visualization in Figure 8 shows that our design performs as expected. BRHVC focuses its attention on more valuable regions, and this mechanism enables it to handle the URC issue more effectively.
>
> Third, experiments demonstrate that BRHVC is capable of handling the URC issue more effectively. The experiments in Table 4 show that BRHVC achieves a significant overall performance improvement. Moreover, the experiments in Table 8 (from Reviewer iau7's Weakness 6) show that BRHVC performs better than DCVC-B when the URC problem is more severe.
>
> **Question 1: Experiment of removing BCF/BMC.**
>
> The newly added Table 4 presents the experimental results for these modules, where *Baseline w/ BCF,BMC* and *Baseline* corresponds to our *BRHVC* and *BRHVC w/o BCF,BMC*, respectively. For details on the baseline design, please refer to Reviewer L376's Weakness 1.
>
> **Question 2: Validate optical flows at different scales.**
>
> The optical flows at each scale contribute to a noticeable performance improvement. To investigate the role at each scale, we designed the following experiment: we iteratively replace the optical flows at one scale with the downsampled/upsampled version from another scale and then fine-tune the network until convergence. The results are in Table 5, where "$(\cdot)\uparrow_2$" denotes 2× upsample, "$(\cdot)\downarrow_2$" denotes 2× downsample. "$v^1, v^2, v^3,\hat{v}^1,\hat{v}^2,\hat{v}^3$" are all shorthand for the corresponding scales: specifically, "$v^1$" denotes $v^1_{t,f}, v^1_{t,b}$, and likewise for the others. For example, Method M1 replaces the largest-size flows $v^1,\hat{v}^1$ with the intermediate-scale flows $v^2,\hat{v}^2$, its result measures the performance change caused by the absence of $v^1,\hat{v}^1$. Likewise, M2 and M3 denote replacing the intermediate- and smallest-size optical flows, respectively.
>
> **Table 5:** Contributions of the optical flows at different scales.
>
> | Method | Encoder Flows | Decoder Flows | BD-rate |
> | --- | --- | --- | --- |
> | BRHVC | --  | --  | 0   |
> | M1  | $v^1\leftarrow (v^2)\uparrow_2$ | $\hat{v}^1 \leftarrow (\hat{v}^2)\uparrow_2$ | 6.7% |
> | M2  | $v^2\leftarrow (v^1)\downarrow_2$ | $\hat{v}^2 \leftarrow (\hat{v}^1)\downarrow_2$ | 4.2% |
> | M3  | $v^3\leftarrow (v^2)\downarrow_2$ | $\hat{v}^3 \leftarrow (\hat{v}^2)\downarrow_2$ | 3.0% |
>
> Experimental results demonstrate that **the flow at every scale provides a noticeable gain**, demonstrating that leveraging additional flows indeed enables more effective motion compensation. Moreover, optical flows at different scales exhibit varying contributions. As the spatial resolution decreases, the contribution of the optical flow at that scale diminishes due to the reduced amount of information it carries.
>
> **Question 3:  Carify the two main B-frame coding strategies.**
>
> Thank you for your feedback. We will specifically distinguish the type of coding in the manuscript.

---

> > ### Comment · Reviewer_BaaM · 2025-08-05
> >
> > Thank you to the authors for the detailed response. The replies did address some of my concerns.
> >
> > (1, 2) Regarding complexity: The increase in kMACs is not negligible, even if it does not directly reflect actual usage; it nevertheless indicates the computational requirements—especially with highly optimized implementations/hardware. However, I agree that the increases in complexity/memory/etc., are within an acceptable range given the performance gains obtained.
> > The only remaining concern I have regarding the comparison is whether the reference model was well trained. Since its increased complexity comes solely from greater depth, it may be difficult to optimize—stacking a very larger number ResBlocks (e.g. 9 blocks * 3 scales for fusion) can be challenging, especially in video‐coding models, which are often shallow. I suggest that the authors provide additional justification for their choice of reference models in the revised paper. For example, increasing the complexity but without such larger number of blocks.
> >
> >
> > (3) For the issue of how existing models fail to use reference information, I still believe the authors’ explanation is not sufficient to support their claim. Although existing works only utilize convolutional layers to combine references, in theory these layers could still learn to leverage two references under different situations. Indeed, designing a mechanism tailored to this issue could be helpful—especially since the network doesn’t always learn perfectly (e.g., due to capacity or data)—but this still does not explain how existing models fail. While Figure 8 demonstrates how the proposed mechanisms work, it does not show how existing models struggle to handle this issue.
> >
> > In my opinion, providing sufficient support for the proposed phenomenon is essential. The authors can’t simply identify a new issue and propose a solution without adequate justification.

---

> > > ### Author Response · Authors · 2025-08-05
> > >
> > > Thank you very much for your detailed comments. Below are our responses to the concerns you raised.
> > >
> > > **Supplementary Question 1: Complexity.**
> > >
> > > We acknowledge that the current *Baseline w/ Resblocks* design may appear overly complex. In response to your concerns, please allow us to offer the following clarifications.
> > >
> > > First, thank you for your agreement that the complexity of our model remains within an acceptable range given the performance gains obtained.
> > >
> > > Second, we confirm that *Baseline w/ Resblocks* was trained with exactly the same stategy as BRHVC and reached convergence successfully. We acknowledge that it remains uncertain whether this training strategy fuly unleashes the potential of the model. However, we can ensure that the comparative methods were trained under the same strategy, thereby quaranteeing relative fairness.
> > >
> > > Third, the large number of Resblock layers was chosen primarily to match BRHVC’s coding speed. Following your suggestions, maybe we can experiment with a lighter model that is constrained to the same kMACs. However, we regret that we have not been able to further design and train such a model, and we kindly ask for your understanding given the limited rebuttal time.
> > >
> > > **Supplementary Question 2: How existing models fail to use reference information.**
> > >
> > > Indeed, as you pointed out, a purely convolutional architecture can learn the appropriate weights in theory. However, previous convolution-based methods model the reference weights implicitly, whereas BRHVC does so explicitly. Please allow us to explain as follows.
> > >
> > > In prior works, the reference weights are in the parameters of the convolutional kernels and are identical across all testing images, ignoring the differences in reference images **(forward & backward)** and especially the hierarchical levels inherent in RA **(level 1 to 5)**. Consequently, the network may fail to explicitly distinguish among image contents and hierarchical structure. Instead, it may rely on the implicit, layer-by-layer processing of standard convolutions to analyse the situation. Although in theory convolution could handle it, it remains unclear whether it can reliably distinguish different content due to its lack of interpretability.
> > >
> > > By contrast, in BRHVC the reference weights are computed by the BCF. This mechanism dynamically adjusts the weights according to image content, supplying the correct references before they are fed into the encoder/decoder. Moreover, it offers meaningful interpretability, as illustrated in Figure 8.
> > >
> > > To explain more clearly, we can think of it this way: just as dynamic convolution [S3] builds on standard convolution by letting the network compute extra weights for every input, we apply the same idea—extra weights that change according to each reference’s contribution.
> > >
> > > [s3] Chen, Yinpeng, et al. "Dynamic convolution: Attention over convolution kernels." In CVPR. 2020.

---

> > > > ### Comment · Reviewer_BaaM · 2025-08-08
> > > >
> > > > Thank you to the authors for their further response.
> > > >
> > > > - I understand that time constraints make it difficult to run additional experiments, but I hope the authors will consider these suggestions into the updated paper to strengthen the experimental design.
> > > >
> > > > - Although the proposed module output is interpretable, it still does not fully address whether existing models fail to learn the same behaviour. I agree that the proposed mechanism is designed in a reasonable way to adapt to different references during coding; however, demonstrating of the limitations of current models would, in my opinion, be an even more valuable contribution than introducing a stronger model, as it would better benefit future models’ design.
> > > >
> > > >
> > > > I will take the authors’ replies into account when preparing my final justification.

---

> > > > > ### Author Response · Authors · 2025-08-08
> > > > >
> > > > > We sincerely thank you for your constructive suggestions, which are very valuable in improving our paper. The following is our response:
> > > > >
> > > > > **Concern 1: More appropriate comparison model.**
> > > > >
> > > > > This suggestion is highly reasonable. We will include experiments in the updated version that compare models of different complexities, as previously mentioned, to provide a more comprehensive evaluation.
> > > > >
> > > > > **Concern 2: How existing models fail to use reference information.**
> > > > >
> > > > > We agree with your perspective. We think it is indeed difficult to prove that existing models cannot learn appropriate reference information, as it is implicitly captured within the convolutional layers, making it difficult to visualize or explicitly confirm whether these models are truly learning the correct reference information.
> > > > >
> > > > > However, based on prior research, we believe that explicit mechanisms are more likely to achieve this goal. Some recent methods [s4, s5] have employed this explicit weighting mechanism in P-frame coding. Moreover, as mentioned in Section 2.2, such explicit designs have been shown to be effective in early B-frame compression studies [45, 47, 7, 14]. Building on them, we extend this mechanism from the pixel domain to the multi-scale feature domain in order to explore its additional performance potential in NBVC. Therefore, we consider that our condition-coding-based explicit reference-weighting method may provide a meaningful reference for future NBVC works.
> > > > >
> > > > > Finally, we will respect your final decision. Once again, we sincerely appreciate the time and expertise you have dedicated to reviewing our work.
> > > > >
> > > > > [s4] Chen, Yi-Hsin, et al. "Maskcrt: Masked conditional residual transformer for learned video compression." *IEEE TCSVT*. 2024.
> > > > >
> > > > > [s5] Liu, Haojie, et al. "End-to-end neural video coding using a compound spatiotemporal representation." *IEEE TCSVT*. 2024.

---

> > > ### Author Response · Authors · 2025-08-08
> > >
> > > Thank you once again for your invaluable comments. In response, we have conducted more detailed comparative experiments and an ablation study on optical flow. These additional results further confirm that our performance gains are justified and that the added complexity remains acceptable. We appreciate your generous acknowledgment of these points.
> > >
> > > We have further provided theoretical analysis of why existing methods require improvement, as you specifically emphasized.
> > >
> > > As the discussion phase draws to a close, we would be grateful if you could confirm whether these revisions adequately address your concerns. We sincerely look forward to your final feedback.

---

### Note · Authors · 2025-08-14

We sincerely thank all reviewers, ACs, and SACs for their time, effort, and expertise on this work. We will incorporate the reviewers' valuable suggestions into the updated version. Finally, whether this paper is accepted or not, we will fully respect the final decision.

---

### Decision · Program_Chairs · 2025-09-17

**Decision:**

Accept (poster)

**Comment:**

This paper introduces BRHVC, a novel framework that addresses the Unbalanced Reference Contribution problem, a significant but underexplored challenge in neural B-frame video compression. Through its innovative Bi-directional Motion Converge (BMC) and Contextual Fusion (BCF) modules, the method achieves SOTA performance, even surpassing the traditional VTM-RA codec on standard datasets. The authors' engagement during the rebuttal was exemplary; they provided extensive new experiments and offered thorough, compelling responses to the reviewers' questions.

However, despite the paper's clear contributions, the initial reviews raised several critical concerns, focusing on the model's computational overhead, comparisons against equivalently complex models, performance under different coding configurations, and the performance drop on specific high-resolution/high-frame-rate datasets. The authors' rebuttal proved decisive: they provided detailed complexity analyses (kMACs/memory), supplied missing ablation studies, demonstrated performance across different IP settings, and reasonably explained the model's limitations on high-resolution video with new downsampling experiments, successfully resolving all core issues.

This productive discussion was directly reflected in the final reviewer stances; all reviewers were satisfied with the authors' responses, and those who were initially "borderline accept" explicitly raised their ratings, forging a strong and unanimous consensus for acceptance. Therefore, given the work's solid technical contributions and the authors' comprehensive resolution of all critical points raised during the review process, my recommendation is to Accept.